# Interaction-Centric Knowledge Infusion and Transfer for Open-Vocabulary Scene Graph Generation

**Lin Li**[1,2], **Chuhan Zhang**[1,2], **Dong Zhang**[1,2], **Chong Sun**[3], **Chen Li**[3], **Long Chen**[1*]

[1]HKUST   [2]AI Chip Center for Emerging Smart Systems   [3] Tencent

{llllidy, chuhanzhang, dongz, longchen}@ust.hk, {waynecsun, chaselli}@tencent.com

https://github.com/HKUST-LongGroup/ACC

## Abstract

Open-vocabulary scene graph generation (OVSGG) extends traditional SGG by recognizing novel objects and relationships beyond predefined categories, leveraging the knowledge from pre-trained large-scale models. Existing OVSGG methods always adopt a two-stage pipeline: 1) *Infusing knowledge* into large-scale models via pre-training on large datasets; 2) *Transferring knowledge* from pre-trained models with fully annotated scene graphs during supervised fine-tuning. However, due to a lack of explicit interaction modeling, these methods struggle to distinguish between interacting and non-interacting instances of the same object category. This limitation induces critical issues in both stages of OVSGG: it generates noisy pseudo-supervision from mismatched objects during knowledge infusion, and causes ambiguous query matching during knowledge transfer. To this end, in this paper, we propose an inter**AC**tion-**C**entric end-to-end OVSGG framework (**ACC**) in an interaction-driven paradigm to minimize these mismatches. For *interaction-centric knowledge infusion*, ACC employs a bidirectional interaction prompt for robust pseudo-supervision generation to enhance the model's interaction knowledge. For *interaction-centric knowledge transfer*, ACC first adopts interaction-guided query selection that prioritizes pairing interacting objects to reduce interference from non-interacting ones. Then, it integrates interaction-consistent knowledge distillation to bolster robustness by pushing relational foreground away from the background while retaining general knowledge. Extensive experimental results on three benchmarks show that ACC achieves state-of-the-art performance, demonstrating the potential of interaction-centric paradigms for real-world applications.

## 1   Introduction

Scene graph generation (SGG) [55] aims to map an image into a structured semantic representation, where objects are expressed as nodes and their relationships are as edges within the graph. Recently, with the burgeoning of large-scale models, *e.g.*, vision-language models (VLMs) and multimodal large language models (MLLMs), OVSGG [14, 29, 9] has emerged as a promising area. It pushes beyond predefined categories to support the recognition and generation of novel objects and relationships, holding great potential for real-world applications.

Generally, an end-to-end VLM-based[2] OVSGG pipeline consists of two phases: **Knowledge Infusion** and **Knowledge Transfer**. The **former** infuses knowledge from large-scale datasets into VLMs via pre-training. This process aims to achieve visual-concept alignment via caption-region comparison. Specifically, due to the lack of region-level information (*e.g.*, bounding box annotations), recent work [14, 63, 9] adopts a weakly-supervised strategy to generate ⟨subject, predicate, object⟩

---

*Long Chen is the corresponding author.

[2]We primarily discuss VLM-based models due to the high resource demands of MLLM-based approaches.

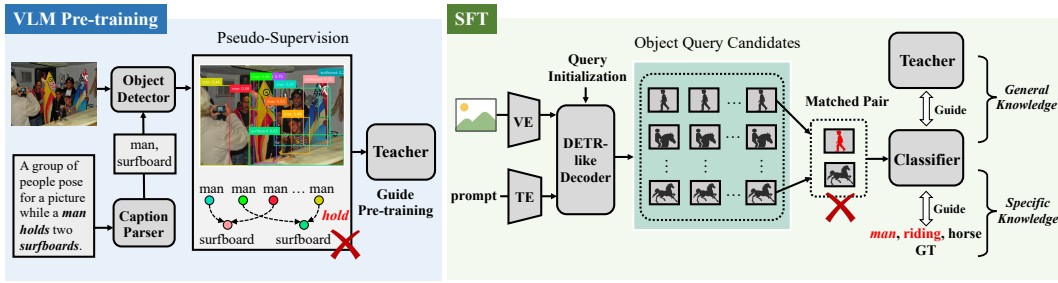

**(a) Knowledge Infusion**          **(b) Knowledge Transfer**

Figure 1: Overview of the end-to-end OVSGG framework challenges. a) Knowledge Infusion, using solely object categories for detection causes ambiguity in associating object pairs (*e.g.*, identifying the correct "`man-surfboard`" for the "`hold`"). b) Knowledge Transfer, vast object query[3] candidates make misaligned non-interacting objects (*e.g.*, "`man` 🧍") with interacting training target "`man`" in ⟨`man`, `riding`, `horse`⟩.

triplets with bounding boxes as pseudo-supervisions. As displayed in Figure 1(a), this framework first extracts semantic graphs from image captions using SGG parsers [41], then grounds objects in the graphs with off-the-shelf object detectors (*e.g.*, Faster R-CNN [39], GLIP [27] and Grounding DINO [35]). The **latter** transfers knowledge from pre-trained VLMs by refining with task-specific objectives and high-quality annotations during supervised fine-tuning (SFT). Concretely, for specific knowledge, it finetunes part of VLM's parameters [9] or adapts prompt-tuning [14] on SGG dataset with fully-supervised triplet annotations (*c.f.* Figure 1(b)). Leveraging these bounding box annotations, a DETR-like structure [4, 66, 62] with bipartite graph matching is typically used to align predicted object queries[3] with ground-truths. Moreover, knowledge distillation (KD) [13, 60, 9] is widely used during SFT, where a generalist VLM (teacher) guides the target model (student) to retain general knowledge, allowing robust adaptation to unseen categories in open-world scenarios.

Despite notable advancements, prevailing OVSGG frameworks exhibit an *object-centric paradigm* in both knowledge infusion and transfer, *i.e.*, lack of interaction differentiation between instances within the same object category. For example, the `man` involved in a holding action and the `man` without any action are represented in an indistinguishable manner. It can amplify ***mismatches in relation pairs*** across both pre-training stage (*i.e.*, knowledge infusion) and SFT stage (*i.e.*, knowledge transfer), which induces the following drawbacks: ❶ *Bringing noisy supervision in pre-training*. As illustrated in Figure 1(a), relying solely on entity categories (*e.g.*, `man` and `surfboard`) to detect objects generates a large number of candidate pairs. This ambiguity makes it hard to associate relation (*e.g.*, "`hold`") to the proper object pair (*e.g.*, "`man-surfboard`"). Using mismatched triplets (*e.g.*, `man` in red and `surfboard` in pink) further exacerbates the confusion, hindering the training of robust SGG models. ❷ *Leading mismatched objects during SFT*. Due to the vast object query candidates, a non-interacting "`man` 🧍" query can be mistakenly associated with `man` in the triplet annotation ⟨`man`, `riding`, `horse`⟩ during bipartite graph matching, as displayed in Figure 1(b). However, the real target is another "`man` 🏇" engaged in riding. This mismatch further complicates the relation classification task, making it harder to predict correct interactions.

In this paper, we propose the inter**AC**tion-**C**entric end-to-end OVSGG framework (**ACC**), which fundamentally rethinks knowledge infusion and transfer through an interaction-driven paradigm. Unlike conventional object-centric approaches that treat all instances uniformly, ACC explicitly models relational dynamics at both pre-training and SFT stages to reduce the pervasive mismatch between interacting/non-interacting object pairs. For **interaction-centric knowledge infusion**, we devise a bidirectional interaction prompt to facilitate visual triplet pseudo-supervision generation, thereby infusing more robust interaction knowledge into pre-trained VLMs. These prompts incorporate interaction tokens that capture contextual dependencies and relational semantics, enabling the grounding model to distinguish interacting objects (*e.g.*, `man` involved in holding action) from non-interacting ones through the attention mechanism [49]. For **interaction-centric knowledge transfer**, to achieve the paradigm shift from object-centric to interaction-centric knowledge transfer, we first establish *interaction-guided query selection*, a two-step mechanism to prioritize interacting objects and incorporate relational context into the query selection process, mitigating interference of

---

[3]Object queries are learnable embeddings input to its Transformer decoder, each specializing through attention to global image features to predict a unique object's localization and classification.

inactive objects and reducing mismatches in bipartite graph matching. To preserve general knowledge, we further incorporate *interaction-consistent KD* to realize both point-wise semantic alignment and inter-pair relational consistency among teacher and student. By explicitly modeling the relative dependencies between interaction-based and non-interaction pairs, this KD strategy enhances the model's robustness in handling novel triplet combinations and background and avoiding catastrophic general knowledge forgetting [13, 9].

To evaluate ACC, we conducted comprehensive experiments on the benchmark Visual Genome (VG) [19], GQA [16], and PSG [56] datasets to validate its effectiveness in addressing the key challenges of OVSGG. In summary, our contributions are threefold:

- We reveal key limitations in existing OVSGG frameworks, *i.e.*, the neglect of interaction-specific characteristics among instances of the same object category during knowledge infusion and transfer, which leads to widespread relation pair mismatches.

- We propose an interaction-centric end-to-end OVSGG framework ACC, shifting the paradigm from object-level representations to interaction-driven learning. By explicitly encoding interactions during both knowledge infusion and transfer, ACC enables more accurate scene graph generation and robust generalization to unseen categories.

- Extensive experiments on three prevalent SGG benchmarks demonstrate the effectiveness and generalizability of ACC.

## 2 Related Work

**Open-Vocabulary SGG (OVSGG).** This task bridges the gap between closed-set SGG and real-world requirements by leveraging VLMs or MLLMs to generalize beyond predefined categories [38, 35, 22, 57]. Current approaches fall into two main groups: 1) *VLM-based Methods.* They primarily rely on contrastive pre-training to align visual and textual embeddings. By comparing visual features of unseen objects/relations and their semantics in common spaces, these models (*e.g.*, CLIP [38] and Grounding DINO [35]) enable zero-shot generalization. Recent advancements, such as He *et al*. [14], explore visual-relation pre-training and prompt fine-tuning for OVSGG. Yu *et al*. [59] leverage CLIP to align relational semantics in multimodal spaces, while Chen *et al*. [9] use a student-teacher framework to improve open-set relation prediction. Besides, other methods integrate class-level descriptions [25, 20] or scene-level descriptions [6] to enrich the semantic context and enhance the discrimination among different relationships. 2) *MLLM-based Methods.* They extend the capabilities of VLMs by incorporating auto-regressive language models, predicting objects and relations in an open-ended manner. Specifically, they use the sequential prediction capabilities of MLLMs, *e.g.*, BLIP [21] and LLaVA [33], to model scene graphs as structured sequences [24]. For example, PGSG [29] and OpenPSG [65] employ auto-regressive models to iteratively predict open-ended objects and relations. ASMv2 [51] builds on LLaVA [33] with instruction tuning [52], unifying both object localization [42, 43] and relation comprehension [23, 26]. Despite their power, MLLM-based methods typically require huge computing resources. This work focuses on VLM-based methods and proposes an interaction-centric framework that explicitly models interactions and enhances generalization to novel categories.

**Knowledge Infusion and Transfer for Open-Vocabulary Learning.** Recent VLM advancements unlock open-vocabulary downstream tasks via two main steps: 1) Knowledge infusion into VLMs (*e.g.*, CLIP [38]) via contrastive learning on large image-text pairs for aligned visual-textual representations. 2) Knowledge transfer by SFT of pre-trained VLMs with task-specific objectives and high-quality annotations for adaptation to tasks like open-vocabulary object detection/segmentation. Within this framework, effectively mining semantic knowledge and leveraging transferable representations has emerged as a key research area to improve generalization in open-world settings while reducing computational/annotation costs. For instance, Wu *et al*. [54] replace the DETR-style encoder with CLIP's visual encoder and employ prompt tuning [30] to adapt image-level representations to region-level tasks for OV object detection. Similarly, Cho *et al*. [10] fine-tune CLIP for open-vocabulary segmentation by incorporating cost aggregation techniques [15]. Besides, Chen *et al*. [9] extend this framework to OVSGG and build upon the GroundingDINO [35] with knowledge distillation to preserve learned knowledge, but still under an object-centric paradigm. Conversely, this work emphasizes interaction-centric knowledge infusion and transfer for robust OVSGG.

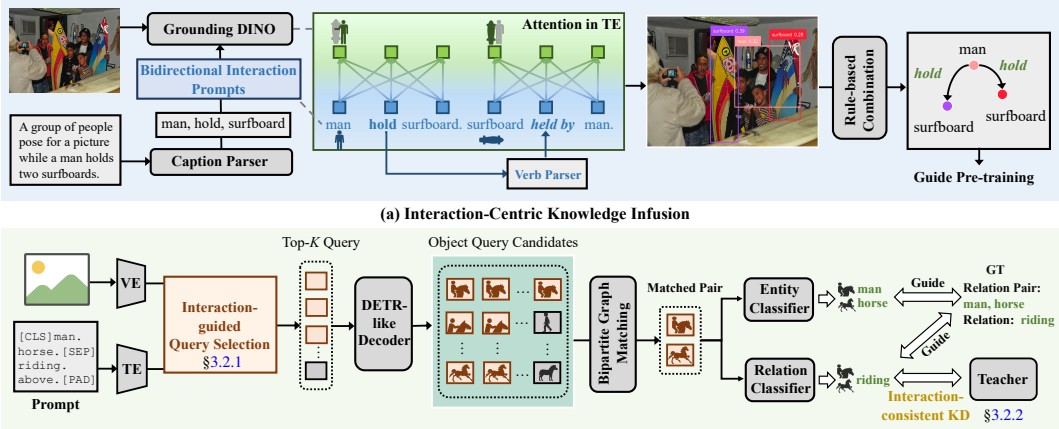

**(a) Interaction-Centric Knowledge Infusion**

**(b) Interaction-Centric Knowledge Transfer**

Figure 2: Overview of ACC for OVSGG. (a) Interaction-Centric Knowledge Infusion: Employs bidirectional interaction prompts and rule-based bounding box combinations for robust pseudo-supervision, empowering the model's grasp of relational knowledge. (b) Interaction-Centric Knowledge Transfer: Uses interaction-guided query selection to prioritize learning on interacting objects, and interaction-consistent KD transfers comprehensive relational insights from the pre-trained VLM to ensure robust generalization to novel categories.

**Knowledge Distillation (KD).** This strategy trains a smaller "student" model to replicate the outputs of a larger "teacher" model, commonly used in open-vocabulary learning to transfer knowledge from VLMs. It encourages the student to mimic the teacher's enriched hidden space, enabling generalization from base to novel concepts. Prior work [13, 60] explores KD in open-vocabulary object detection by using L1/MSE loss to align the student detector's features with the teacher VLM's regional visual features. However, this hard alignment may fail to capture complex feature structures. Later work [2, 37] aligns the similarity of inter-embeddings, aiding in the acquisition of structured knowledge. Recent work extends to multi-scale level [50] or bags-of-region level [53], contrasting with InfoNCE loss. This paper adopts an interaction-consistent KD that combines point-to-point concept retention and structure-aware interaction retention distillation, preserving teacher's knowledge and identifying novel relationships beyond backgrounds.

## 3  ACC: Interaction-Centric End-to-end OVSGG Framework

**Formulation.** Given an image $I$, SGG aims to construct a structured semantic graph $\mathcal{G} = (\mathcal{V}, \mathcal{E})$. Each node $v_i \in \mathcal{V}$ is defined by its bounding box (bbox) and category, while each edge $e_{ij} \in \mathcal{E}$ represents the relationship between $v_i$ and $v_j$. In **open-vocabulary settings**, the label set $\mathcal{C}$ for nodes and edges is divided into *base classes* $\mathcal{C}_B$ and *novel classes* $\mathcal{C}_N$, such that $\mathcal{C}_B \cup \mathcal{C}_N = \mathcal{C}$ and $\mathcal{C}_B \cap \mathcal{C}_N = \emptyset$. $\mathcal{C}_B$ contains seen classes during training, while $\mathcal{C}_N$ includes unseen classes that the model is expected to generalize to during inference.

**Baseline End-to-End OVSGG Architecture.** As illustrated in Figure 2(b), an end-to-end OVSGG framework [9, 36] typically follows a dual-encoder-single-decoder architecture [35], involving three main components:

- **Visual and Text Encoders.** Visual encoder (VE) extracts multi-scale visual features $\mathbf{V} \in \mathbb{R}^{N_v \times d}$. Text encoder (TE) processes textual prompts that concatenate all predefined object and relation categories [63, 9], *e.g.*, "[CLS] man. horse. [SEP] riding. above. [PAD]" to derive semantic embeddings for objects $\mathbf{T}_o \in \mathbb{R}^{N_o \times d}$ and relation $\mathbf{T}_r \in \mathbb{R}^{N_r \times d}$. Here, $N_v$, $N_o$, and $N_r$ denote the numbers of image, object, and relation tokens, respectively. $d$ is the feature dimension.

- **DETR-like Decoder.** It refines the representations of $K$ object queries $\{\mathbf{q}_i\}_{i=1}^{K}$ through self-attention and cross-attention mechanisms, leveraging both visual and text features, ultimately predicting object bounding box coordinates. Besides, a global relation query $\mathbf{q}_{rel}$ is often utilized to capture spatial and semantic dependencies among objects [44, 9].

- **Entity and Relation Classifiers.** For open-vocabulary recognition, node features $\{\mathbf{e}_o\}$ (derived from refined object queries) and edge features $\{\mathbf{e}_{ij}\}$ (constructed by combining features of paired

objects, potentially augmented with global relation embeddings) are compared against the textual object/relation class embeddings.

The training of such models conventionally relies on *bipartite graph matching* to align object queries with ground-truth annotations, minimizing a cost function based on semantic and spatial criteria [4]. Optimization objectives usually include bbox regression loss (L1 $\mathcal{L}_{reg}$ and GIoU loss $\mathcal{L}_{giou}$ [40]), cross-entropy based entity and relation classification losses ($\mathcal{L}_{obj}$ and $\mathcal{L}_{rel}$)[4]. However, the efficacy of this process is undermined if the supervision is noisy (due to object-centric pre-training) or if query-to-target alignment is confounded by non-interacting distractors (due to object-centric SFT). To surmount the limitations imposed by current end-to-end OVSGG designs, ACC introduces a fundamental shift towards an *interaction-driven* paradigm. As illustrated in Figure 2, ACC incorporates interaction-centric knowledge infusion and transfer.

### 3.1 Interaction-Centric Knowledge Infusion

Addressing the challenge of noisy supervision from object-centric pseudo-labeling (issue ❶ in §1), ACC's knowledge infusion stage fundamentally alters how training targets are generated for VLM pre-training. To ensure that pseudo-supervision effectively captures interaction distinctiveness, especially within weakly annotated data, ACC conditions the object detection process on interactional context rather than relying on prompts based on isolated object classes (*e.g.*, "`man. surfboard.`").

To be specific, after the semantic graph parsing process, which extracts initial subject-predicate-object triplets from image captions with a language parser [41], we employ Grounding DINO [35] as the object detector and design **bidirectional interaction prompt** to guide the object localization. The bidirectional interaction prompt is constructed by combining two perspectives for each interaction triplet: one reflecting the action from the subject's viewpoint (*e.g.*, "`man hold surfboard`") and another from the object's perspective (*e.g.*, `surfboard held by man`"). The former is directly derived from the components of the interaction triplet, while the latter converses the subject and object with a *counter-action* (*e.g.*, "`held by`") generated by the verb parser. This verb parser is typically an LLM[5] (*e.g.*, Llama2 [48] and Qwen [1]) or Python Library.

The dual-perspective construction process brings two key advantages: 1) *Modeling Context Information*: Through the attention mechanism in the text encoder of Grounding DINO, the bidirectional interaction prompt integrates contextual interaction information into object tokens. As shown in Figure 2(a), the attention mechanism enables the token "`man`" to absorb relevant interaction semantics, such as "`hold surfboard`", ensuring that the grounded object "`man`" is correctly aligned with its interaction context. 2) *Enhancing Object Role Awareness*: By reversing operation, the object (*e.g.*, "`surfboard`") of given triplet becomes the syntactic subject of the whole sentence (*e.g.*, "`surfboard held by man`"). As the central of the rephrased sentence, the syntactic subject receives heightened attention, improving its accuracy in localization.

Furthermore, inspired by [31, 17], we adopt a *rule-based combination* that combines overlapping subject and object bounding boxes to form triplet supervision by Intersection over Union (IoU) score.

### 3.2 Interaction-Centric Knowledge Transfer

The interaction-centric knowledge infused during pre-training (§3.1) provides a strong foundation. Nevertheless, it still faces a mismatch problem during SFT (issue ❷ in §1). ACC's interaction-centric knowledge transfer is designed to ensure that 1) the selection and refinement of object queries are explicitly guided by interaction potential, and 2) the rich, interaction-focused knowledge from the pre-trained model is adequately transferred and further enhanced to discriminate between genuine interactions and non-interacting background. This is achieved through interaction-guided query selection and interaction-consistent knowledge distillation.

#### 3.2.1 Interaction-Guided Query Selection

To mitigate mismatched object pairs during SFT, interaction-guided query selection instills an interaction prior into the two-step query generation process to reduce non-interacting candidates.

---

[4]Detailed formulations are left in appendix §B.

[5]The generation process of counter-action is in the appendix §C.

**Step I.** This step aims to directly identify the most relevant visual tokens likely to participate in object interactions. Intuitively, the visual features of interacting objects should exhibit strong correlations with both object and relation semantics. To achieve this, for each visual token $\mathbf{v}_i \in \mathbf{V}_v$, a relevance score $s_i$ is computed by combining its maximum similarity with object and relation class tokens:

$$s_i = \left( \max(\mathbf{v}_i \mathbf{T}_o^\top) \right)^\gamma \cdot \left( \max(\mathbf{v}_i \mathbf{T}_r^\top) \right)^{1-\gamma}, \tag{1}$$

where $\max(\mathbf{v}_i \mathbf{T}_o^\top)$ computes the maximum similarity between the visual token $\mathbf{v}_i$ and all object class tokens in $\mathbf{T}_o$, while $\max(\mathbf{v}_i \mathbf{T}_r^\top)$ computes the maximum similarity between $\mathbf{v}_i$ and all relation class tokens in $\mathbf{T}_r$. The parameter $\gamma \in [0,1]$ balances their contributions. Based on the relevance scores, the top $K$ query indices, denoted as $\mathcal{I}_K$, are selected by the following procedure:

$$\mathcal{I}_K = \text{Top}_K(\{s_i \mid i = 1, 2, \ldots, N_v\}). \tag{2}$$

The visual features and the position embedding [62, 35] corresponding to the selected indices $\mathcal{I}_K$ are used to initialize queries for further decoding operations.

**Step II.** Nevertheless, Step I's individual encoding of object and relation tokens struggles to capture deeper interaction semantics and distinguishes among objects. Thus, Step II explicitly models interaction semantics by integrating relational context into the object tokens. Specifically, after the initial forward pass, the model predicts a set of visual relation triplets. These triplets are decomposed into interaction pairs ⟨`subject`, `predicate`⟩ and ⟨`predicate`, `object`⟩, which serve as *interaction-guided prompts*. These prompts are encoded via the TE of VLM to get interaction tokens embeddings $\mathbf{T}_{in}$. The decomposition process has dual advantages: First, the predicates within prompts guide the TE's attention to infuse object tokens with interaction information, enabling the model to capture contextual dependencies and enhance its understanding of relationships. For instance, the token "`man`" can incorporate the semantic meaning of the interaction "`riding`" to obtain "`man` 🐎" in Figure 2(b). Second, decomposing triplets into pairs avoids direct interference between object tokens, effectively preserving their unique characteristics. As illustrated in Figure 2(b), "`man` 🐎" and "`horse` 🐎" are independently processed, preventing unnecessary dependencies across unrelated categories and maintaining the individual semantics of each object.

For each visual token $\mathbf{v}_i$, the interaction relevance score $s_i^{in}$ is calculated by measuring the maximum similarity with interaction tokens:

$$s_i^{in} = \max(\mathbf{v}_i \mathbf{T}_{in}^\top). \tag{3}$$

The query indices set prioritizes the top $L$ tokens with the highest interaction relevance:

$$\mathcal{I}_L^{in} = \text{Top}_L(\{s_i^{in} \mid i = 1, 2, \ldots, N_v\}). \tag{4}$$

However, relying solely on interaction relevance may fail to identify objects absent from the initially predicted triplets yet crucial for comprehensive scene understanding. To address this, the object relevance score $s_i^o$ is computed similarly, but using object tokens $\mathbf{T}_o$. The remaining $K - L$ query indices are selected based on object relevance, excluding those already chosen:

$$\mathcal{I}_{K-L}^o = \text{Top}_{K-L}(\{s_i^o \mid i \notin \mathcal{I}_L^{in}, i = 1, 2, \ldots, N_v\}). \tag{5}$$

The final query indices set combines these two subsets $\mathcal{I}_K = \mathcal{I}_L^{in} \cup \mathcal{I}_{K-L}^o$. This two-step strategy effectively reduces non-interacting candidates and mitigates bipartite graph mismatches. Pseudo-code detailing this process is in appendix §D for clarity.

### 3.2.2 Interaction-Consistent Knowledge Distillation

Beyond localization and classification objectives, we adopt interaction-consistent KD to enhance the model's ability to distinguish interacting pairs from background and address catastrophic forgetting of learned relational semantics [9]. Specifically, it uses the VLM pre-trained in the first stage as the teacher model. The student network is designed as a pseudo-siamese structure of the teacher, initialized with its parameters.

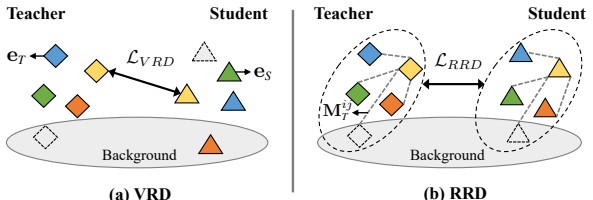

Figure 3: Illustration of interaction-consistent KD.

This KD combines *visual-concept retention distillation* (VRD) and *relative-interaction retention distillation* (RRD) to align the student model with the teacher's semantic space while maintaining inter-pair relational consistency. The entire loss function contains two complementary objectives:

**VRD.** The first objective draws from [9] ensures that the student's edge features remain point-wise consistent with the teacher's semantic space for negative samples. The loss is defined as:

$$\mathcal{L}_{VRD} = \frac{1}{|\mathcal{N}|} \sum_{\mathbf{e} \in \mathcal{N}} \|\mathbf{e}_S - \mathbf{e}_T\|_1, \tag{6}$$

where $\mathbf{e}_S$ and $\mathbf{e}_T$ represent student and teacher's edge features, and $\mathcal{N}$ is the set of negative samples.

**RRD.** While VRD effectively preserves point-wise semantic consistency, it fails to ensure the relative relationships between triplets, *i.e.*, distinguishing interaction pairs from backgrounds (*c.f.* Figure 3(a)). RRD explicitly models inter-pair relativity [2, 37] by aligning the structure similarity of triplet embeddings between the teacher and student models. The structure similarity matrices for the teacher and student models, $\mathbf{M}_T$ and $\mathbf{M}_S$, are normalized by L2 norm:

$$\mathbf{M}_T^{ij} = \frac{\mathbf{e}_T^i \cdot \mathbf{e}_T^{j\top}}{\|\mathbf{e}_T^i \cdot \mathbf{e}_T^{j\top}\|_2}, \quad \mathbf{M}_S^{ij} = \frac{\mathbf{e}_S^i \cdot \mathbf{e}_S^{j\top}}{\|\mathbf{e}_S^i \cdot \mathbf{e}_S^{j\top}\|_2}. \tag{7}$$

RRD then aligns these similarity matrices by minimizing the Frobenius norm $\| \cdot \|_F$ between them:

$$\mathcal{L}_{RRD} = \frac{1}{|\mathcal{N}|^2} \|\mathbf{M}_S - \mathbf{M}_T\|_F^2. \tag{8}$$

**Final Objectives:** Combine localization and classification losses with above complementary objectives to achieve point-wise semantic alignment and relational consistency:

$$\mathcal{L} = \mathcal{L}_{reg} + \mathcal{L}_{giou} + \mathcal{L}_{obj} + \mathcal{L}_{rel} + \beta_1 \mathcal{L}_{VRD} + \beta_2 \mathcal{L}_{RRD}. \tag{9}$$

The weights $\beta_1$ and $\beta_2$ control the importance of semantic alignment and relational consistency.

## 4 Experiments

### 4.1 Experiment Setup

**Datasets.** We evaluated ACC on three SGG benchmarks: 1) **VG** [19] contains annotations for 150 object categories and 50 relation categories across 108,777 images. Following standard setup [55], 70% of the images are used for training, 5,000 for validation, and the remaining for testing. For a fair comparison, we excluded images overlapping with the pre-training dataset of Grounding DINO [35], retaining 14,700 test images as in [63]. 2) **GQA** [16] uses the GQA200 split [12, 45], including 200 object categories and 100 predicate categories. We randomly sampled 70% of the object and predicate categories as the base, and more details can be found in the appendix §A. 3) **PSG** [56] offers 44,967 training, 1,000 test, and 3,000 validation images (sampled from training), with 133 object and 56 predicate categories. We adopted the same base and novel class splitting in [29].

**Settings.** Following [9], we compared ACC under two settings: 1) **OvR-SGG**: Evaluates generalization to unseen relations while retaining original object categories. Fifteen of 50 relation categories in VG150 are removed during training, with performance measured on "Base+Novel (Relation)" and "Novel (Relation)". 2) **OvD+R-SGG**: Assesses handling of unseen objects and relations simultaneously. Both novel objects and relations are excluded during training, evaluated on "Joint Base+Novel", "Novel (Object)", and "Novel (Relation)".

**Metrics.** We conducted experiments under the challenging Scene Graph Detection (**SGDET**) protocol [55, 19], which requires detecting objects and identifying relationships between object pairs without GT object labels or bounding boxes. We reported: 1) **Recall@K** (**R@K**): The proportion of ground-truth triplets correctly predicted within the top-K confident predictions. 2) **Mean R@K** (**mR@K**): The average R@K across all categories.

**Implementation Details.** Due to space constraints, details are provided in the appendix §A.

### 4.2 Comparison with State-of-the-Art Methods

We compared ACC with existing state-of-the-art methods, *i.e.*, **VS**³ [63], **OvSGTR** [9], and **RAHP** [36]. The experimental results on the VG dataset [19] under both the OvR-SGG and OvD+R-SGG setups are shown in Table 1 and Table 2, respectively. Notably, ACC consistently outperforms the latest SOTA methods across all metrics. In the OvR-SGG setup, ACC surpasses the RAHP (Swin-T) by **+1.78**% R@100 within the novel relation categories, demonstrating superior generalization

Table 1: Experimental results of OvR-SGG setting on VG [19] test set.

| Method | Backbone | Base+Novel (Relation) | | | Novel (Relation) | | |
|---|---|---|---|---|---|---|---|
| | | R@20 | R@50 | R@100 | R@20 | R@50 | R@100 |
| IMP [55] CVPR'17 | - | - | 12.56 | 14.65 | - | 0.00 | 0.00 |
| MOTIFS [61] CVPR'18 | - | - | 15.41 | 16.96 | - | 0.00 | 0.00 |
| VCTREE [47] CVPR'19 | - | - | 15.61 | 17.26 | - | 0.00 | 0.00 |
| TDE [46] CVPR'20 | - | - | 15.50 | 17.37 | - | 0.00 | 0.00 |
| OpenSGen [18] ICMR'25 | - | - | 18.00 | 20.50 | - | 15.70 | 17.90 |
| VS$^3$ [63] CVPR'23 | | - | 15.60 | 17.30 | - | 0.00 | 0.00 |
| OvSGTR [9] ECCV'24 | Swin-T | - | 20.46 | 23.86 | - | 13.45 | 16.19 |
| RAHP [36] AAAI'25 | | - | 20.50 | 25.74 | - | 15.59 | 19.92 |
| **ACC (Ours)** | | **17.49** | **23.22** | **27.40** | **12.90** | **17.89** | **21.70** |
| OvSGTR [9] ECCV'24 | Swin-B | - | 22.89 | 26.65 | - | 16.39 | 19.72 |
| **ACC (Ours)** | | **18.77** | **24.81** | **29.28** | **14.72** | **20.04** | **24.66** |

Table 2: Experimental results of OvD+R-SGG setting on VG [19] test set.

| Method | Backbone | Joint Base+Novel | | | Novel (Obj) | | | Novel (Rel) | | |
|---|---|---|---|---|---|---|---|---|---|---|
| | | R@20 | R@50 | R@100 | R@20 | R@50 | R@100 | R@20 | R@50 | R@100 |
| IMP [55] CVPR'17 | - | - | 0.77 | 0.94 | - | 0.00 | 0.00 | - | 0.00 | 0.00 |
| MOTIFS [61] CVPR'18 | - | - | 1.00 | 1.12 | - | 0.00 | 0.00 | - | 0.00 | 0.00 |
| VCTREE [47] CVPR'19 | - | - | 1.04 | 1.17 | - | 0.00 | 0.00 | - | 0.00 | 0.00 |
| TDE [46] CVPR'20 | - | - | 1.00 | 1.15 | - | 0.00 | 0.00 | - | 0.00 | 0.00 |
| VS$^3$ [63] CVPR'23 | | - | 5.88 | 7.20 | - | 0.00 | 0.00 | - | 0.00 | 0.00 |
| OvSGTR [9] ECCV'24 | Swin-T | 10.02 | 13.50 | 16.37 | 10.56 | 14.32 | 17.48 | 7.09 | 9.19 | 11.18 |
| **ACC (Ours)** | | **12.61** | **17.43** | **21.27** | **12.48** | **17.16** | **21.10** | **11.38** | **15.90** | **19.46** |
| OvSGTR [9] ECCV'24 | Swin-B | 12.37 | 17.14 | 21.03 | 12.63 | 17.58 | 21.70 | 10.56 | 14.62 | 18.22 |
| **ACC (Ours)** | | **13.50** | **18.88** | **23.19** | **13.46** | **18.84** | **23.29** | **12.37** | **17.50** | **21.73** |

and reduced overfitting. With the Swin-B backbone, ACC achieves 29.28% R@100, which is higher than OvSGTR across both base and novel relations, further emphasizing its robustness. In the more challenging OvD+R-SGG scenario, ACC continues to outperform the competition. Specifically, on the joint base and novel classes, ACC gains **+4.90**% and **+2.16**% R@100 over OvSGTR with the Swin-T and Swin-B backbones, respectively. These results validate ACC's superior performance and robust generalization across both relation and object domains.

## 4.3 Diagnostic Experiment

To ensure a comprehensive evaluation, we performed a series of ablation studies on the VG dataset [19] in the challenging OvD+R-SGG scenario. More experimental analyses are left in the appendix.

**Knowledge Infusion Part.** We analyzed the effectiveness of ACC's bidirectional interaction prompt (BIP) for pseudo-supervision generation (§3.1) in Table 3. It can be seen that BIP leads to consistent improvements across all metrics. Notably, when compared to the configuration without BIP, it achieves R@100 gains of **1.73**% on the joint base and novel classes, **1.45**% on

Table 3: Ablation studies (§4.3) on BIP.

| Method | Split | R@20 | R@50 | R@100 |
|---|---|---|---|---|
| **Ours** | Joint Base+Novel | **12.61** | **17.43** | **21.27** |
| *w/o* BIP | | 11.84 | 16.17 | 19.55 |
| **Ours** | Novel (Obj) | **12.48** | **17.16** | **21.10** |
| *w/o* BIP | | 12.36 | 16.09 | 19.65 |
| **Ours** | Novel (rel) | **11.38** | **15.90** | **19.46** |
| *w/o* BIP | | 10.73 | 14.40 | 17.83 |

novel object classes, and **1.63**% on novel relation classes, respectively. This demonstrates that BIP effectively improves performance by considering interaction contexts in supervision generation.

**Knowledge Transfer Part.** We evaluated the efficacy of ACC's interaction-guided query selection (IGQS §3.2.1) and interaction-consistent KD (ICKD §3.2.2) in the knowledge transfer phase. The re-

Table 4: Ablation studies (§ 4.3) on IGQS and ICKD.

| Components | | Joint Base+Novel | | | Novel (Obj) | | | Novel (Rel) | | |
|---|---|---|---|---|---|---|---|---|---|---|
| IGQS | ICKD | R@20 | R@50 | R@100 | R@20 | R@50 | R@100 | R@20 | R@50 | R@100 |
| | | 10.02 | 13.50 | 16.37 | 10.56 | 14.32 | 17.48 | 7.09 | 9.19 | 11.18 |
| ✓ | | 11.37 | 15.71 | 19.37 | 11.43 | 15.80 | 19.61 | 9.84 | 13.92 | 17.38 |
| | ✓ | 11.43 | 15.67 | 19.20 | 11.57 | 15.65 | 19.32 | 10.07 | 14.00 | 17.32 |
| ✓ | ✓ | 11.84 | 16.17 | 19.55 | 12.36 | 16.09 | 19.65 | 10.73 | 14.40 | 17.83 |

sults are summarized in Table 4, with the first row representing the baseline OVSGG pipeline with *visual-concept retention distillation* from [9]. From this analysis, three key conclusions can be drawn: **First**, IGQS refines the query selection process. By prioritizing interacting objects and minimizing mismatched assignments, IGQS achieves notable improvements, such as **3.00**% R@100 gains, highlighting its ability to enhance precision by focusing on interacting object pairs. **Second**,

Table 5: Comparison with pre-training methods. All models are **pre-trained** on image-caption data and tested on VG150 [19] test set directly. Our models trained on COCO captions are used as pre-trained models.

| SGG model | Backbone | Grounding | R@20 | R@50 | R@100 |
|---|---|---|---|---|---|
| LSWS [58] CVPR'21 | - | - | - | 3.28 | 3.69 |
| MOTIFS [61] CVPR'18 | - | Li *et al.* [31] | 5.02 | 6.40 | 7.33 |
| Uniter [8] ECCV'20 | - | SGNLS [64] | - | 5.80 | 6.70 |
| Uniter [8] ECCV'20 | - | Li *et al.* [31] | 5.42 | 6.74 | 7.62 |
| VS$^3$ [63] CVPR'23 | | GLIP-L [27] | 5.59 | 7.30 | 8.62 |
| OvSGTR [9] ECCV'24 | Swin-T | Grounding DINO [35] | 6.61 | 8.92 | 10.90 |
| **ACC (Ours)** | | Grounding DINO [35] | **7.86** | **10.81** | **13.31** |
| OvSGTR [9] ECCV'24 | Swin-B | Grounding DINO [35] | 6.88 | 9.30 | 11.48 |
| **ACC (Ours)** | | Grounding DINO [35] | **8.28** | **11.61** | **14.33** |

leveraging interaction-consistent KD with *relative-interaction retention distillation* ensures relational consistency during training, resulting in significant performance boosts. It contributes **2.83**% R@100 gains, improving the model's ability to handle novel classes effectively. **Third**, the integration of two components yields the best overall performance, with **1.80**%∼**6.65**% improvements across all evaluation metrics. However, the improvement is less pronounced than expected, since each strategy prioritizes interacting objects, which may lead to diminishing returns by progressively reducing non-interacting objects. Despite this, the combined results still demonstrates enhanced relational understanding and serve as a valuable tool for improving performance in complex scenarios.

**Supervision Analysis.** We investigated ACC's impact in the pre-training process (*c.f.* Table 5). As seen, models pre-trained on COCO [7] captions with ACC variants consistently outperform others, achieving **13.31**% R@100 with Swin-T and **14.22**% R@100 with Swin-B. These results demonstrate the effectiveness of ACC in the VLM pre-training process.

In addition, we visualized the object detection results from ACC and the original methods that solely use object categories for detection. As displayed in Figure 2, the original method produces redundant objects, complicating the identification of subject-object interactions. For instance,

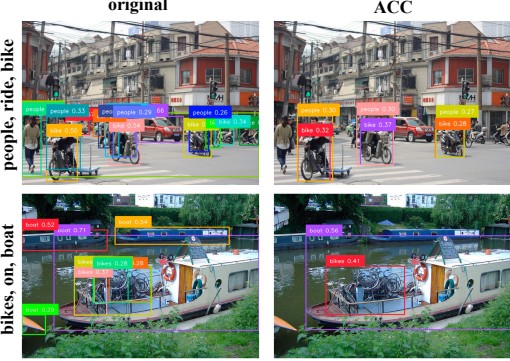

Figure 4: Pseudo supervision generation in ACC.

given the "⟨people, ride, bike⟩" triplet, the baseline detects multiple instances of "people" and "bike", obscuring the interaction. In contrast, ACC leverages bidirectional interaction prompts and attention mechanisms to accurately localize the interaction-relevant objects. A similar enhancement is observed in the "⟨bikes, on, boat⟩" triplet, where ACC focuses on interaction-relevant entities.

**Query Visualization.** To demonstrate the effectiveness of IGQS, we visualized the top-50 selected queries in Figure 5. As seen, the original approach makes no distinction between instances within the same category, such as "man" or "zebra", resulting in both interacting and non-interacting instances receiving a similar number of queries. This indiscriminate query generation increases the likelihood of incorrect matches during bipartite graph matching, as irrelevant regions compete with interaction-relevant instances. Conversely, IGQS prioritizes interacting queries ("man holding" or "zebra laying on" in Figure 5), increasing discrimination among objects with the same categories.

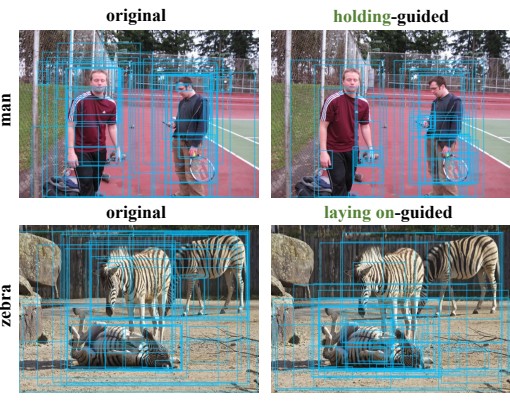

Figure 5: Interaction-guided query selection.

## 5   Conclusion

This work presented ACC, an interaction-centric OVSGG framework. ACC alleviates current paradigms' failure to distinguish interacting from non-interacting instances by adopting interaction-

centric principles in two key phases. Knowledge infusion uses a bidirectional interaction prompt for robust pseudo-supervision, enhancing interaction understanding; knowledge transfer combines interaction-guided query selection with interaction-consistent knowledge distillation to mitigate mismatches and irrelevant object interference. ACC shows significant improvements on three main benchmarks. We anticipate that ACC will not only set new standard for OVSGG but also inspire further exploration of interaction-driven strategies in VLMs for more accurate scene understanding.

**Acknowledgement.** This work was supported by the National Natural Science Foundation of China Young Scholar Fund (62402408). This research was partially conducted by ACCESS – AI Chip Center for Emerging Smart Systems, supported by the InnoHK initiative of the Innovation and Technology Commission of the Hong Kong Special Administrative Region Government. This research was also supported by the Hong Kong SAR RGC Early Career Scheme (26208924) and sponsored by Tencent WeChat Rhino-Bird Focused Research Program.

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

## Summary of the Appendix

To facilitate a deeper understanding of the main paper, we present supplementary material with additional details, organized as follows:

- §A elaborates on the implementation details.
- §B introduces the formulations of training objectives.
- §C introduces the counter-action generation prompt.
- §D provides the pseudo-code for interaction-guided query selection.
- §E offers additional experimental results.
- §F presents further qualitative results.
- §G discusses our limitations, broader impact, and directions of future work.

## A   Implementation Details

**Pre-training.** Our models are trained with a batch size of 3, utilizing four/eight RTX 3090 GPUs for computation. During the supervision generation phase (*c.f.* §3.1), we employ Llama2-7B [48] to generate counter-actions based on the prompts described in §C. A pseudo-triplet class is annotated when the confidence of the grounding object class exceeds 0.25, and the intersection over union (IoU) between the subject and object is greater than 0.0. During pre-training, we initialize our model using the pre-trained Grounding DINO checkpoints provided by [9], keeping the visual backbone (Swin-T or Swin-B) and the text encoder (BERT-base [11]) frozen. The remaining modules, such as the relation-aware embedding, are initialized randomly. In line with [9], we select 100 object detections per image for pairwise relation recognition during training.

**Supervised Fine-Tuning.** The supervised fine-tuning process is conducted using the same computational resources as pre-training. For interaction-guided query selection (*c.f.* §3.2.1), we adopt the settings from [35, 9], where the total number of selected visual tokens $K$ is set to 900, and the top-ranked interaction tokens $L$ is fixed at 200. The models after the pre-training process are leveraged as the teacher model and serve as the initialization for the student model. The weights $\beta_1$ and $\beta_2$ of the loss function $\mathcal{L}_{VRD}$ and $\mathcal{L}_{RRD}$ are set to 0.1 and 0.5, respectively, to balance different optimization objectives.

**Dataset Splits.** All entity and relation categories for the GQA dataset [16] are listed in Table S1. For the VG dataset [19], we adopted the splitting protocol from [9]. As for the PSG dataset [56], we followed the splits utilized in [29].

## B   Training Objectives

As mentioned in §3, the model is guided by the bounding box regression loss, entity classification loss, and relation classification loss. This section details their corresponding formulations.

**Bounding Box Regression Loss**: The primary objective of object localization is to accurately predict the positions and sizes of objects within an image. To achieve this, the model utilizes a combination of L1 loss ($\mathcal{L}_{reg}$) and GIoU loss ($\mathcal{L}_{giou}$) [40], ensuring both precise positioning of the bounding boxes and effective handling of overlaps. The corresponding loss functions are defined as:

$$\mathcal{L}_{reg} = \frac{1}{N_b} \sum_{i=1}^{N_b} \|\hat{\mathbf{b}}_i - \mathbf{b}_i\|_1,$$

$$\mathcal{L}_{giou} = 1 - \frac{A_{inter}}{A_{union}} + \frac{(A_{min} - A_{union})}{A_{min}}, \tag{10}$$

where $\hat{\mathbf{b}}_i$ and $\mathbf{b}_i$ denote the predicted bounding box and GT bounding box, respectively. $N_b$ is the number of the object's bounding boxes. $A_{inter}$ represents the area of intersection between the predicted and ground truth bounding boxes, $A_{union}$ is the union area of the bounding boxes, and $A_{min}$ is the area of the smallest enclosing box covering both.

**Entity Classification Loss**: To address the class imbalance in the object classification task, the model employs Focal Loss ($\mathcal{L}_{obj}$) [32], which emphasizes difficult-to-classify and underrepresented

Table S1: The categories spitting of GQA [16].

| Split | Relation Categories | Object Categories |
|---|---|---|
| Base | parked on, growing on, standing in front of, wearing, standing on, with, looking at, under, carrying, near, above, covered in, behind, at, using, hanging from, sitting on, flying in, watching, covering, mounted on, in front of, lying on, standing next to, grazing in, holding, beside, on the back of, catching, running on, swimming in, playing on, on top of, floating in, talking on, on the bottom of, standing behind, leaning against, covered by, facing, filled with, attached to, sitting next to, next to, worn on, in, on the side of, driving, close to, surrounded by, lying in, hitting, pulling, swinging, touching, eating, throwing, skiing on, driving on, hang on, riding, playing in, crossing, walking with, on, growing in, sitting in, cutting, feeding, leaning on | mountain, cow, people, face, number, pizza, tire, player, pillow, screen, truck, kite, trunk, sock, neck, glove, coat, letter, roof, windshield, desk, paw, leaf, flower, plant, counter, paper, eye, book, branch, lamp, cup, phone, toilet, skateboard, logo, laptop, vehicle, motorcycle, hill, curtain, nose, sheep, bowl, wire, bear, banana, mouth, drawer, shelf, cap, animal, bottle, box, airplane, finger, room, flag, seat, tower, wing, fruit, rock, house, pot, bird, umbrella, surfboard, lady, tie, fork, vase, bag, orange, clock, sidewalk, food, sink, cabinet, beach, boat, basket, helmet, child, racket, post, guy, towel, arm, napkin, bush, bench, person, cone, apple, jacket, fur, air, sign, bus, wrist, frame, floor, dress, street, shoe, ball, girl, ear, boy, broccoli, fence, uniform, hair, sneakers, blanket, zebra, train, camera, sticker, license plate, lid, tomato, pants, giraffe, watch, wall, leg, bed, t-shirt, shorts, horse, spots, arrow, field, bread, bicycle, knife, couch, ceiling |
| Novel | on the front of, reaching for, flying, of, parked along, talking to, sitting at, standing by, hanging on, covered with, standing near, full of, surrounding, walking in, reflected in, walking down, walking on, contain, below, printed on, driving down, waiting for, resting on, playing with, standing in, grazing on, by, around, pulled by, beneath | ocean, car, picture, hand, snow, horn, woman, sweater, container, paint, feet, clouds, foot, dirt, faucet, chair, sand, tail, stone, cat, tag, traffic light, keyboard, tree, leaves, elephant, ground, glass, frisbee, trash can, word, man, jeans, door, building, sky, table, wheel, pole, collar, hat, cheese, mane, shirt, dog, cord, cake, donut,plate, backpack, mirror, street light, skis, window, grass, water, bike, road, head, cell phone |

categories. Focal Loss modifies the standard cross-entropy loss by down-weighting easy examples, focusing the model's attention on challenging ones. The formulation is as follows:

$$\mathcal{L}_{obj} = -\alpha(1 - y_c)^\gamma \log(y_c), \tag{11}$$

where $y_c$ denotes the predicted probability of the true object class $c$, $\alpha$ is a balancing factor, and $\gamma$ is a focusing parameter that adjusts the emphasis on hard examples.

**Relation Classification Loss**: The model's objective is to predict the relationships between objects, aligning predicted relation scores with ground-truth annotations. This is achieved using BCE loss ($\mathcal{L}_{rel}$), which measures the discrepancy between predicted and true relationship probabilities. The BCE loss function is expressed as:

$$\mathcal{L}_{rel} = -\frac{1}{N_{rel}} \sum_{i=1}^{N_{rel}} \left[ y_{ij} \log(\hat{y}_{ij}) + (1 - y_{ij}) \log(1 - \hat{y}_{ij}) \right], \tag{12}$$

where $y_{ij}$ denotes the GT relation label between the $i$-th object and $j$-th object, $\hat{y}_{ij}$ is the predicted relation probability.

## C   Counter-action Generation Prompt

In this section, we present the prompt used for counter-action generation (*c.f.* §3.1) in Figure S1 with LLMs, *i.e.*, Llama2 [48]. The prompt is structured into two key components: the example and the question. **Example:** The example, such as the instance of "ride", serves as a reference for the model to produce contextually relevant outputs in an in-context learning framework [3, 34]. This part of the prompt is also generated by the LLM, providing a model-driven demonstra-tion of the expected output format. **Question:** The question, *i.e.* "please generate...", prompts the model to produce a corresponding counter-action or a related output. This structure ensures that the

> **Question:** Given the action 'ride', please generate its corresponding counter-action.
> **Answer:** 'be ridden by'.
> **Question:** Given the action 'eat', please generate its corresponding counter-action.
> **Answer:** 'be eaten by'.
> ________________________________________
> **Question:** Given the action '{**relation**}', please gener-ate its corresponding counter-action.
> **Answer:**

Figure S1: Counter-action generation prompt.

**Algorithm S1** Pseudo-code for Step I in Interaction-Guided Query Selection.

```
X_v: visual features of all tokens.
X_o: object class tokens.
X_r: relation class tokens.
gamma: balancing parameter.

def Step1_QuerySelection(X_v, X_o, X_r, gamma):
    scores = []
    for i in range(len(X_v)):
        # Calculate the relevance score for each visual token.
        sim_o = max(X_v[i] @ X_o.T) # max similarity with object class tokens
        sim_r = max(X_v[i] @ X_r.T) # max similarity with relation class tokens
        score = (sim_o ** gamma) * (sim_r ** (1 - gamma)) # Eq. (1)
        scores.append(score)

    # Select top K visual tokens based on relevance score.
    I_K = top_K(scores, K)

    return I_K
```

**Algorithm S2** Pseudo-code for Step II in Interaction-Guided Query Selection.

```
X_v: visual features of all tokens.
X_in: interaction tokens from text encoder.
X_o: object class tokens.

def Step2_QuerySelection(X_v, X_in, X_o):
    interaction_scores = []
    for i in range(len(X_v)):
        # Compute interaction relevance score based on interaction tokens.
        sim_in = max(X_v[i] @ X_in.T) # max similarity with interaction tokens
        interaction_scores.append(sim_in)

    # Select top L visual tokens based on interaction relevance score.
    I_L_in = top_L(interaction_scores, L)

    # Compute object relevance for remaining tokens.
    object_scores = []
    for i in range(len(X_v)):
        if i not in I_L_in:
            sim_o = max(X_v[i] @ X_o.T) # max similarity with object class tokens
            object_scores.append(sim_o)

    # Select top (K-L) tokens based on object relevance.
    I_K_L_o = top_K_minus_L(object_scores, K - L, I_L_in)

    # Final query set is the union of both sets.
    I_K = I_L_in + I_K_L_o

    return I_K
```

model can generate responses specific to the action at hand, supporting more relevant and consistent counter-action generation.

# D   Pseudo Code

To make the interaction-guided query selection (§3.2.1) process easier to understand, we provide pseudo-code for Step I and Step II in Algorithm S1 and Algorithm S2, respectively.

# E   More Experimental Results

## E.1   Comparison with State-of-the-Arts on GQA dataset

In Table S2, we compared our ACC with the existing SOTA method (*i.e.*, OvSGTR [9]) on the GQA [16] dataset under the more challenging OvD+R-SGG setting. The backbones are uniformly set to Swin-T. Notably, ACC consistently outperforms OvSGTR across all metrics, demonstrating the universality and effectiveness of our approach.

Table S2: Experimental results of OvD+R-SGG setting on GQA [16] test set.

| Method | Joint Base+Novel | | | Novel (Obj) | | | Novel (Rel) | | |
|---|---|---|---|---|---|---|---|---|---|
| | R@20 | R@50 | R@100 | R@20 | R@50 | R@100 | R@20 | R@50 | R@100 |
| OvSGTR [9] ECCV'24 | 11.21 | 15.80 | 19.14 | 10.32 | 14.92 | 18.76 | 2.59 | 5.21 | 7.40 |
| **ACC (Ours)** | **12.30** | **16.88** | **20.63** | **11.51** | **16.16** | **20.57** | **3.41** | **6.60** | **9.80** |

Table S3: Experimental results of OvR-SGG setting on PSG [56] test set.

| Method | Joint Base+Novel | | | Novel (Rel) | | |
|---|---|---|---|---|---|---|
| | R@20 | R@50 | R@100 | R@20 | R@50 | R@100 |
| SGTR [28] CVPR'22 | - | 14.2 | 18.2 | - | - | - |
| PGSG [29] CVPR'24 | - | 18.0 | 20.2 | - | - | - |
| OvSGTR [9] ECCV'24 | 15.14 | 17.76 | 19.50 | 5.32 | 6.93 | 8.08 |
| **ACC (Ours)** | **16.69** | **20.01** | **21.71** | **6.78** | **8.78** | **9.70** |

Table S4: Extra metrics of OvD+R-SGG setting on VG150 [19] test set.

| Method | Base (Obj) | | | Base (Rel) | | | Novel (Obj) | | | Novel (Rel) | | |
|---|---|---|---|---|---|---|---|---|---|---|---|---|
| | R@20 | R@50 | R@100 | R@20 | R@50 | R@100 | mR@20 | mR@50 | mR@100 | mR@20 | mR@50 | mR@100 |
| OvSGTR [9] | 8.78 | 11.95 | 14.79 | 12.07 | 16.47 | 20.09 | 1.69 | 2.44 | 3.06 | 0.82 | 1.13 | 1.47 |
| **ACC (Ours)** | 11.66 | 16.46 | 20.35 | 12.20 | 16.67 | 20.57 | 1.93 | 2.84 | 3.61 | 1.64 | 2.59 | 3.38 |

## E.2 Comparison with State-of-the-Arts on PSG dataset

Given that the PSG dataset split proposed by [29] exclusively addresses novel relation categories, our evaluation consequently focused on the OVD-R-SGG setting. As detailed in Table S3, when compared with other prominent state-of-the-art methods (*e.g.*, SGTR [28], PGSG [29], and OvSGTR [9]), our ACC framework also demonstrates superior performance across all reported metrics.

## E.3 Evaluation with More Metrics

We reported both recall of base classes and mean Recall (m@R) in S4. It can be seen that our ACC outperforms the previous SOTA method (*i.e.*, OvSGTR [9]) in both metrics. This demonstrates that our approach provides a more comprehensive and powerful generalization capability, enhancing performance across the board, not just for unseen classes.

## E.4 Ablation Study on Interaction-Centric Knowledge Infusion

**Effectiveness of bidirectional interaction prompt**. To investigate the bidirectional interaction prompt's sensitivity to the choice of verb parser for counter-action generation, we replaced the de-

Table S5: Ablation study on the verb parser in counter-action generation.

| Method | Verb Parser | Size | Joint Base+Novel | | |
|---|---|---|---|---|---|
| | | | R@20 | R@50 | R@100 |
| **ACC (Ours)** | Llama2 | 7B | 13.50 | 18.88 | 23.19 |
| **ACC (Ours)** | Qwen2.5 | 0.5B | 13.64 | 18.99 | 23.43 |
| **ACC (Ours)** | Pattern (Python Lib) | - | 13.36 | 18.56 | 22.64 |

fault Llama2 parser with two alternatives: a smaller Large Language Model (LLM), Qwen2.5-0.5B [1], and the Pattern (a Python library) under OvD+R-SGG setting on VG test set (Swin-B as backbone). As shown in Table S5, ACC sustains high performance even when utilizing a smaller LLM or a non-LLM parser for this task. This demonstrates the robustness of our bidirectional interaction prompt in generating effective pseudo-supervision across various verb parsing mechanisms.

## E.5 Ablation Study on Interaction-Centric Knowledge Transfer

**Effectiveness of query selection.** We performed an ablation study of the two-step query selection in IGQS (*c.f.* §3.2.1), as shown in Table S6. The general end-to-end OVSGG pipeline with visual-concept retention distillation as the baseline. The results demonstrate that using a single step also yields performance improvements over the baseline, with the best performance achieved by employing both steps simultaneously.

Table S6: Ablation on two-step query selection in IGQS on OvD+R-SGG setting of VG150 [19] test set.

| IGQS | | Joint Base+Novel | | |
|---|---|---|---|---|
| Step I | Step II | R@20 | R@50 | R@100 |
| | | 10.02 | 13.50 | 16.37 |
| ✓ | | 11.30 | 15.71 | 19.16 |
| | ✓ | 11.32 | 15.70 | 19.29 |
| ✓ | ✓ | 11.37 | 15.71 | 19.37 |

Table S7: Experimental results of HICO-DET [5] dataset under the OvR-SGG setting.

| Method | Joint Base+Novel | | | Novel (Rel) | | |
|---|---|---|---|---|---|---|
| | R@20 | R@50 | R@100 | R@20 | R@50 | R@100 |
| OvSGTR [9]   ECCV'24 | 34.62 | 37.39 | 39.04 | 22.94 | 28.48 | 31.84 |
| **ACC (Ours)** | 35.74 | 38.58 | 40.19 | 24.44 | 30.77 | 34.38 |

Table S8: Inference costs on the VG150 [19] test set.

| Method | Training costs (min) | Inference costs (s/I) |
|---|---|---|
| OvSGTR [9]   ECCV'24 | 68 | 0.3871220016479492 |
| ACC | 71 | 0.3896771125793457 |
| ACC w/ Step II | 94 | 0.6402182579040527 |

**Hyperparameters in ICKD.** We conducted an ablation study on the hyperparameters ($\beta_1$ and $\beta_2$) in the ICKD (visual-concept retention distillation and relative-interaction retention distillation). Results in Figure S2 show that increasing $\beta_1$ (*e.g.*, raising VRD weight) decreases overall performance, consistent with results

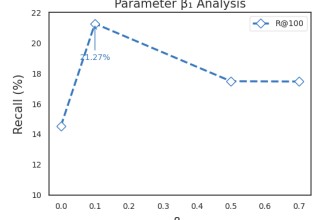 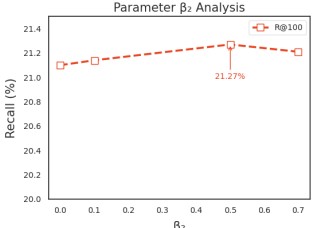

Figure S2: Ablation on $\beta_1$ and $\beta_2$ in VRD and RRD loss function under OvD+R-SGG setting on VG test set.

in [9]. RRD demonstrates robustness for different hyperparameters. The best performance can be achieved with $\beta_1$ = 0.1 and $\beta_2$ = 0.5, respectively.

## E.6  Comparison on Human-Object Interaction Detection Tasks

Human-Object Interaction (HOI) detection, particularly on benchmarks like HICO-DET [5], is primarily a detection task over a set of specific human-centric interactions (*i.e.*, <action, object> pairs). In contrast, SGG addresses a more general and compositional challenge: generating <subject, action, object> triplets between any pair of objects. To empirically validate the effectiveness of the proposed ACC, we evaluated ACC and OvSGTR [9] on the HICO-DET benchmark. As shown in Table S7, ACC consistently outperforms OvSGTR, achieving 2.54% absolute improvement in R@100 of novel classes. This result is significant: it demonstrates that our model's core principles are so robust. They excel not only on the general OVSGG task they were designed for, but also on the specialized HOI task.

## E.7  Computational Overhead

We conducted a time analysis on VG [19], with training on the entire dataset and testing on 20 images. We report the mean value in Table S8 of our ACC (w/o and with Step II in IGQS). We would like to claim that: 1) Our Step I in IGQS just introduces minor computational complexity in elementary matrix operations (*c.f*., Eq. 1). 1) Due to the requirement of forward prediction for self-enhancement, Step II will induce extra computational overhead, but the performance gain brought by Step II is optional.

## F  More Qualitative Comparison Results

### F.1  Grounded Entity Visualization

To evaluate the effectiveness of the proposed bidirectional interaction prompt (§3.1), we visualized the entity grounding results for various types of prompts during the pre-training process in Figure S3:

**Object Prompt.** Prior methods [9] often rely on concatenating the subject and object entities extracted from a scene graph parser, such as "`man. knife`".

**Interaction Prompt.** It incorporates relation triplets into a phrase, *e.g.*, "`man hold knife`."

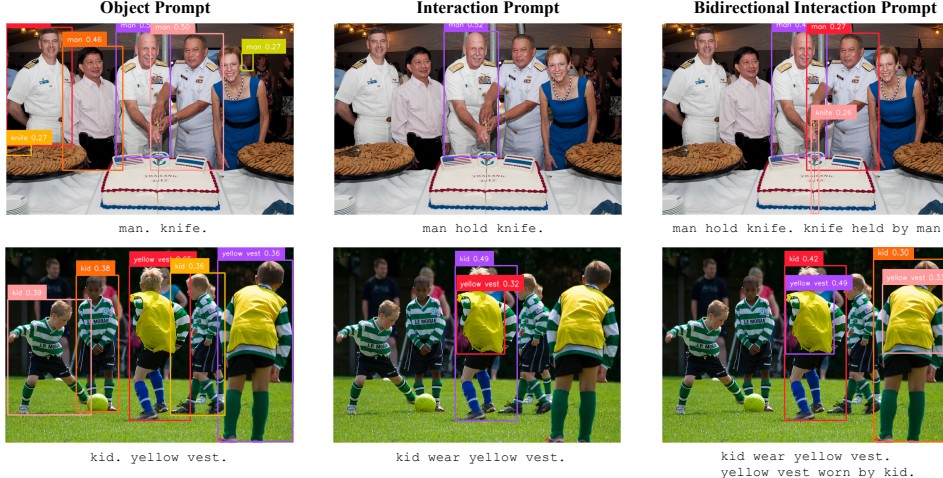

Figure S3: Entity grounding results of different prompts.

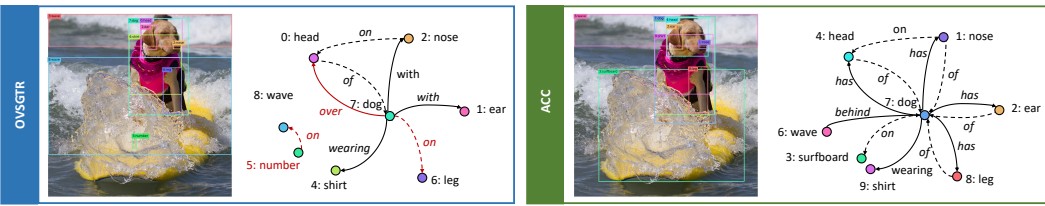

Figure S4: Qualitative Results of OvSGTR and ACC on the SGDet task and VG dataset [19]. The dashed line represents the predicted novel categories, and the **red** represents the unreasonable predictions.

**Bidirectional Interaction Prompt.** This proposed prompt further integrates relation triplets with their corresponding counter-action, forming phrases like "`man hold knife. knife held by man.`".

From the visualization results, the following observations can be made: 1) Directly adopting object prompts tends to generate redundant bounding box candidates (*e.g.*, multiple instances of "`man`" and "`kid`" in Figure S3). This redundancy complicates the identification of interacting object pairs. Additionally, some interacting object boxes are missing. For example, the imperceptibly held "`knife`" is not detected, while the non-interacting "`knife`" is identified. These limitations result in mismatched relational pairs, which ultimately mislead the subsequent training process. 2) While incorporating interaction prompts significantly reduces the number of redundant object boxes, it often over-focuses on the subject (*e.g.*, detecting only the "`man`" subject bounding box), leading to the omission of critical object boxes. 3) By leveraging bidirectional interaction prompts, both the subject and object bounding boxes are accurately detected under the given relation triplet. This approach not only resolves the redundancy issue but also ensures the inclusion of subtle yet crucial interactions (*e.g.*, correctly identifying the "`knife`" being held). Consequently, it provides a more comprehensive and precise grounding for subsequent training stages.

## F.2 Mismatched Examples Visualization.

To intuitively illustrate the challenges stemming from current paradigms, Figure S5 visualizes representative examples of mismatched relational triplets. As depicted, a common error involves triplets such as ⟨man, riding, horse⟩ being incorrectly assigned. This type of mis-

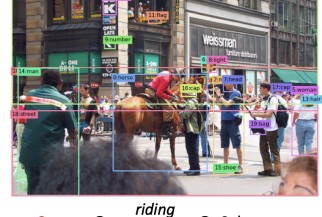
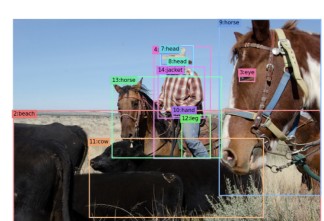

Figure S5: Mismatched relation triplets examples.

assignment frequently occurs be-
cause models lacking explicit interaction modeling struggle to distinguish the specific "`man`" instance
actively engaged in the "`riding`" interaction from other, non-interacting "`man`" instances that may be
present in the scene. Such visualizations highlight the critical need for interaction-centric approaches
to achieve more precise relation recognition.

### F.3 Scene Graph Visualization

Figure S4 displays the scene graph predictions generated by OvSGTR and ACC on the Swin-T
backbone using the VG dataset. Apparently, the scene graph produced by OvSGTR includes several
incorrect and redundant relationships, such as "⟨`number, on, wave`⟩" and "⟨`dog, on, leg`⟩". Instead,
our ACC eliminates such unreasonable predictions and can generate easily missing the relationship
triplet, such as "⟨`wave, behind, dog`⟩". Even interactive relationships, like "⟨`dog, has, head`⟩" and
"⟨`head, of, dog`⟩", are accurately captured, showcasing ACC's enhanced capacity to reason over
subject-object interactions and identify precise and semantically coherent relationships in complex
scenes.

### F.4 Failure Cases Analysis

We analyzed the examples in Fig-
ure S6 where ACC misidentifies non-
interacting object pairs, and find that:
1) For Interaction-Centric Knowledge
Infusion, it is difficult to correctly
match small objects (*e.g.*, `hat` in
background) and their related ob-
jects through bidirectional interaction
prompts. 2) For Interaction-Centric

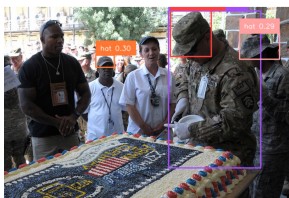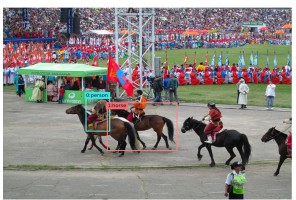

man wear hat. hat worn by man.      person riding horse

Figure S6: Failure cases.

Knowledge Transfer, when multiple subject-object pairs with the same relational triplet categories
(*e.g.*, ⟨`person, riding, horse`⟩) appear in the same image, the model might mistakenly match the
subject in one triplet to the object in another triplet.

## G Discussion

**Limitation Analysis.** Our approach employs a knowledge infusion and transfer framework §2
for open-vocabulary scene graph generation. While this framework reduces annotation costs and
effectively leverages transferable representations from pre-trained vision-language models, it also
inherits inductive biases from the teacher model. Like two sides of a coin, any biases in the vision-
language model toward specific feature traits or classes may propagate to our model. Besides, our
method can alleviate mismatched relational pairs, but cannot avoid all mismatches.

**Potential Broader Impact.** This paper presents work aimed at advancing the field of open vocabulary
scene graph generation. By introducing interaction-aware mechanisms, our approach enhances the
model's ability to recognize novel objects and relationships, improving the robustness and accuracy of
scene understanding in real-world applications such as robotics, autonomous systems, and augmented
reality. While our work has the potential to drive innovation in these fields, ethical considerations
must be taken into account, particularly regarding the fairness and representativeness of the training
data used. Ensuring that our models are inclusive and minimize bias will be crucial to preventing
harmful misinterpretations or exclusions in practical applications.

**Future Work.** Our current algorithm is tailored for open-vocabulary scene graph generation, adopting
a dual-encoder-single-decoder architecture as proposed in [9, 36]. It prioritizes base-novel generaliza-
tion over real-time performance, which may not fully meet the timeliness requirements of real-world
applications. In future work, we aim to enhance the computational efficiency of our approach to
better address these practical demands.