# OpenReview forum: "Interaction-Centric Knowledge Infusion and Transfer for Open Vocabulary Scene Graph Generation"
_NeurIPS.cc/2025/Conference — NeurIPS 2025 poster_

### Official Review · Reviewer_E7mD · 2025-07-02

**Clarity:** 4
**Significance:** 4
**Originality:** 4
**Rating:** 5
**Confidence:** 4

**Summary:**

This paper aims to solve Open Vocabulary Scene Graph Generation (OvSGG) task, where novel objects and their relationships are predicted by SGG models. This paper shifts from object-centric to interaction-centric paradigm in OVSGG. The authors proposes bidirectional interaction prompts and interaction-guided query selection to learn subtle visual feature differences between interacting vs. non-interacting instances within the same object category.

**Questions:**

- Can you incorporate the effect of bidirectional prompt training in generalization study? It seems that the effectiveness can be demonstrated by directly comparing the performance of GLIP and GroundingDINO. I would like to see the effect in the object detection-level, not the relationship-prediction-level.

- If we train the student model with the fully annotated SGG dataset, how can we obtain the open vocab prediction ability? Does the model predict the relationships in the training labels better? (Compared to unseen classes.)

**Ethical Concerns:**

["NO or VERY MINOR ethics concerns only"]

**Final Justification:**

After the authors' rebuttal phase, my concerns about the ambiguous point on distillation paradigm have been addressed. Moreover, I have the positive view on this paper, as the proposed ACC improves the generalization of ability of pre-trained Vision Language Model using interaction-centric model training. During the rebuttal phase, authors provided the experiment about the effectiveness of bidirectional prompt on object detection task, demonstrating the broad impact of this paper not only for visual relationship detection, but also the general object detection task. Hence, I am inclined to accept this paper.

**Limitations:**

While overall motivation is novel and the proposed method is simple and effective, there are some ambiguous points in the paper. For example, the distillation paradigm is difficult to understand. Why is it necessary for OvSGG.

**Paper Formatting Concerns:**

No Concern

**Quality:**

4

**Strengths And Weaknesses:**

## Strength

**Potential for Universal Grounding Enhancement.** Learning the interaction-aware visual representation is very important and technically sound. Current grounding methods like GLIP and GroundingDINO has restricted ability of awarding the interactions. The proposed method, bidirectional interaction prompt, could be a fundamental technique to improve all vision-language grounding models (GLIP, Grounding DINO, etc.) beyond just SGG. While the paper focuses on SGG, the approach has broader applicability that wasn't fully explored.

**Clear Problem Formulation.** The authors explicitly models subtle but crucial interaction context differences like "man holding surfboard" vs "man just standing". Solving the problem through learned representations with new prompts is very novel idea.

**Promising Experimental Result.** The experimental results show that using bidirectional interaction prompt for GroundingDINO greatly improves OvSGG performances on both base and novel relationship classes, achieving 2-5% increases of mR@K. Moreover, the proposed method ACC is fully open vocabulary SGG method, which not only generates unseen relationships but also predicts unseen objects during training, which is demonstrated by the OvD+R-SGG experiment. This implies that ACC is a general important approach that understands the scene without any predefined classes for objects and relationships.

## Weakness

**The necessity of ICKD is difficult to understand.** I cannot catch why student model distillation is necessary since we can train the original models with VRD. The advantage of introducing student model should be clarified in terms of the open vocabulary setting.

---

> ### Author Rebuttal · Authors · 2025-07-31
>
> We deeply appreciate reviewer E7mD for the valuable time and constructive feedback. We provide point-to-point responses below.
>
> **Q1: Generalization study of bidirectional prompt training.**
>
> Thank you for your valuable suggestion. To validate the effect of our bidirectional prompt on the model's generalization ability, we conduct a direct ablation study at the object detection level. The results in **Table_R 12** clearly show that using our bidirectional prompt leads to a significant improvement across all key AP metrics. We will incorporate these results into the revised manuscript.
>
> **Table_R 12: Object detection results on the VG dataset.**
>
> | **Bidirectional prompt** |  AP  | AP50 | AP75 | APs | APm | APl |
> | --- | --- | --- | --- | --- | --- | --- |
> | ✕ | 17.91 | 29.20 | 18.17 | 7.19 | 14.92 | 23.63 |
> | ✓ | 19.46 | 31.75 | 19.72 | 7.79 | 16.20 | 25.42 |
>
> **Q2: Student model with the fully annotated SGG dataset.**
>
> Thank you for the helpful comments. We would like to clarify the following points:
>
> - **Open-Vocabulary Ability.** This capability originates from the model's fundamental **vision-language** architecture. Unlike a traditional classifier with a fixed output category, our model works by computing a **similarity score between visual features and the text embedding** of ***any*** potential relation. This enables it to perform zero-shot predictions for “novel” classes that were never seen during training.
> - **Performance of Base Classes.** Yes, if fine-tuned on the SGG dataset with only standard SGG losses (w/o KD), the model would indeed learn the “base” relations in the training set very well compared to unseen classes (cf. **row 2 in Table_R 13**). But this comes at a huge cost: the model would **catastrophically forget** its general knowledge, leading to a severe collapse in its ability to recognize “novel” classes. This overfitting to base classes is the central challenge in OvSGG, leading directly to why the teacher-student distillation paradigm is not just helpful, but necessary for OvSGG (cf. **row 3 in Tabel_R 13**).
>
> **Table_R 13: Ablation study on fully SGG annotation and ICKD under OvR-SGG setting.**
>
> | Fully annotated SGG dataset | ICKD | Base (Rel) R@20 | Base (Rel) R@50 | Base (Rel) R@100 | Novel (Rel) R@20 | Novel (Rel) R@50| Novel (Rel) R@100|
> | --- | --- | --- | --- | --- | --- | --- | --- |
> | ✕ | - | 1.65 | 2.43 | 3.29 | 12.32 | 16.87 | 20.67 |
> | ✓ | ✕ | 25.41 | 32.56 | 36.68 | 0.91 | 1.61 | 2.34 |
> | ✓ | ✓ | 20.28 | 25.94 | 30.12 | 12.90 | 17.89 | 21.70 |
>
> **Q3: The necessity of ICKD.**
>
> Thank you for this question. We would like to clarify that **VRD** also follows a typical **teacher-student** paradigm, which does not operate on a single model in isolation:
>
> - **The Teacher Model** is from the large-scale pre-training (knowledge infusion). It possesses rich, generalizable knowledge that can be applied to **novel categories**.
> - **The Student Model** is initialized with the teacher's parameters, and it is then fine-tuned on the fully-annotated SGG dataset (knowledge transfer).
>
> In the open-vocabulary setting, if we were to fine-tune the pre-trained model directly (i.e., w/o a teacher's guidance), it would easily overfit to the “base classes” in the SGG dataset. This would severely degrade or even erase its ability to recognize “novel categories”—this is **catastrophic forgetting**. The teacher-student distillation framework acts as a regularizer. The teacher guides the student, forcing its feature space to remain consistent with the teacher's general feature space even as it learns the new task. This preserves the invaluable generalization ability. Within this teacher-student framework, we devise ICKD to optimize the knowledge transfer process:
>
> - **VRD,** ensures the student retains the teacher's understanding of “background” relations (i.e., negative samples), which is fundamental for retaining general knowledge.
> - **RRD,** not only transfers knowledge but also teaches the student the “**relative structure”** among these background relations. This helps the student build a more structured concept of the background, enabling it to more sharply distinguish true novel interactions (the foreground) from the background.
>
> We have performed an ablation study to provide the definitive proof. As shown in **Table_R 13** and **Sec. E.4 in the Appendix,** when our ICKD is removed, performance on novel relations collapses. This empirically demonstrates that distillation is the critical component that prevents the model from overfitting to base classes, thereby unlocking its true and effective open-vocabulary generalization power.

---

> > ### Comment · Reviewer_E7mD · 2025-08-06
> >
> > Thanks for authors' response. Now, I fully understood the rationales of all components in ACC. Moreover, I am satisfied with the object detection result of bidirectional prompt. Hence, I would keep my original positive score on this paper.

---

> > > ### Author Response · Authors · 2025-08-06
> > > **Thanks for Reviewer E7mD's Response**
> > >
> > > We are delighted that you found our response satisfactory. Thank you for your precious time and positive assessment of our work. We will add the relevant discussion to the revision.

---

> ### Comment · Area_Chair_4LJq · 2025-08-05
>
> Dear Reviewer E7mD,
>
> The author-rebuttal phase is now underway, and the authors have provided additional clarifications and performance results in their rebuttal. Could you please take a moment to review their response and engage in the discussion? In particular, we’d appreciate your thoughts on whether their revisions adequately address your initial concerns. Thank you for your time and valuable contributions.
>
> Best, Your AC

---

### Official Review · Reviewer_9mXA · 2025-07-03

**Clarity:** 4
**Significance:** 3
**Originality:** 3
**Rating:** 5
**Confidence:** 3

**Summary:**

The paper introduces ACC — an interACtion-Centric open-vocabulary scene graph generation (OVSGG) framework to enhance the OvSGTR open‐world scene graph detector by adding three interaction‐centric modules to reduce mismatched object–relation pairs: (1) Bidirectional interaction prompt: For each ⟨subject, predicate, object⟩ triplet, the paper generates “object → subject” (“surfboard held by man”) prompts via an LLM to deepen interaction semantics in addition to the “subject → object” (e.g. “man holds surfboard”) interaction. (2) Interaction‐guided query selection: The paper scores predicted visual tokens against object and relation class embeddings — keeping only high‐relevance queries and enriching them with contextual relation cues. (3) Interaction‐consistent knowledge distillation: A student–teacher distillation that not only transfers relational knowledge but also explicitly separates interacting object pairs from background noise.

**Questions:**

1. What is the performance of the model in comparison to the baselines on out-of-distribution data?
2. How much additional computational overhead does ACC incur compared to OvSGTR?
3. Could you please report the results that were promised (mR@K, std. deviation) but seem to be absent from the paper?
4. Could you illustrate using examples or statistics where ACC misidentifies non-interacting object pairs and point to which scene contexts or relation types most frequently lead to those errors?
5. Additional recent baselines for comparison would strengthen the paper.

If the authors address these points (with particular emphasis on points 1 to 3), I would be happy to raise my score unless grave concerns are being brought up by other reviewers.

**Ethical Concerns:**

["NO or VERY MINOR ethics concerns only"]

**Final Justification:**

The authors have addressed most of my concerns as well as those of the other reviewers. I do not see any remaining critical concerns, hence I am raising my score by one notch. That said, my score should be read as somewhere between 4 and 5 as do believe that that the sum of all the added experiments and clarifications especially regarding the motivation are quite substantial. This means that the final paper will need to be rather different than the paper we reviewed. Whether this is too big of a change, I will leave up to the AC to decide, also in comparison to other papers.
Overall, while my assessment is positive, it is not a completely clear case given the substantial amount of added information at the rebuttal stage. If the paper is accepted, I count on the authors that all promised additions will be incorporated.

**Limitations:**

Yes

**Quality:**

3

**Strengths And Weaknesses:**

Strengths:
* The paper is well motivated, very well written, and the results look promising.
* The technical contributions address timely and relevant issues in scene graph generation for improving relation prediction in multimodal LLMs for real-world deployment. I particularly like the bidirectional interaction prompt, which is simple yet rather effective.
* An evaluation on three benchmark datasets shows that ACC consistently outperforms the OvSGTR baseline. Careful ablations show the benefits of the individual components.

Weaknesses:
* The paper claims (l. 284f) that it reports the mR@K metric, which would be quite important to assess long-tail capabilities. but I could not find mR@K numbers in the tables of the main paper or the appendix.
* Similarly, the answer to question 7 in the checklist (l. 659) claims that standard deviations are being reported, but I could not find these in any of the tables.
* Incorporating contextual information likely increases the computational overhead, yet the paper doesn’t discuss this despite claiming otherwise in the checklist (question 8). The appendix only mentions which GPUs where used, but as far as I could see not how long training and inference take.
* Only three recent baselines are being compared with in Tables 1 and 2. The remaining baselines are 5+ years old. More comparisons with recent baselines would clearly strengthen the paper.
* It would be insightful to test ACC versus OvSGTR on out-of-distribution data (e.g., PSG or Open Images) to further evaluate generalization.
* The key notations introduced in the methodology are not defined in Figure 2, making it harder to connect the text to the overall pipeline.

Minor points:
* "ACC achieves R@100 over OvSGTR..." - incomplete sentence.
* Figure 5 is quite interesting. Having quantitative results that support this analysis would make the paper stronger.
* Reference [39] is broken.

---

> ### Author Rebuttal · Authors · 2025-07-31
>
> We deeply appreciate reviewer 9mXA for the valuable time and constructive feedback. We provide point-to-point responses below.
>
> **Q1: Missing mR@K metric.**
>
> Our apologies. We have reported both recall and mean Recall in **Table_R 7**. It can be seen that our ACC outperforms the previous SOTA method (i.e., OvSGTR) in both metrics.
>
> **Table_R 7:  Experimental results of OvD+R-SGG setting on the VG dataset**.
>
> | Method |  Joint Novel + Base R@20/50/100 | Joint Novel + Base mR@20/50/100 | Novel (Obj) R@20/50/100 | Novel (Obj) mR@20/50/100 | Novel (Rel) R@20/50/100 | Novel (Rel) mR@20/50/100 |
> | --- | --- | --- | --- | --- | --- | --- |
> | OvSGTR (Swin-T) | 10.02 / 13.50 / 16.37 | 1.81 / 2.54 / 3.15 | 10.56 / 14.32 / 17.48 | 1.69 / 2.44 / 3.06 | 7.09 / 9.19 / 11.18 | 0.82 / 1.13 / 1.47 |
> | ACC (Swin-T) | 12.61 / 17.43 / 21.27 | 2.09 / 3.02 / 3.80 | 12.48 / 17.16 / 21.10 | 1.93 / 2.84 / 3.61 | 11.38 / 15.90 / 19.46 | 1.64 / 2.59 / 3.38 |
>
> **Q2: Standard deviations.**
>
> Thanks for your keen observation. We have repeated the main experiments five times, and the results (**Table_R 8**) demonstrated high robustness across runs — the standard deviations range from 0.01 to 0.03. Due to the consistently low variance, we initially omitted the standard deviations from the main tables for clarity. However, we acknowledge that this information should have been included in the appendix. In the final version, we will add a dedicated table in the appendix, reporting both the mean and standard deviation for all key results.
>
> **Table_R 8: The 𝐾-times experiment of ACC in OVD+R SGG setting (Joint Base+Novel).**
>
> | No. | R@20 | R@50 | R@100 |
> | --- | --- | --- | --- |
> | 1 | 12.61 | 17.43 | 21.27 |
> | 2 | 12.59 | 17.37 | 21.27 |
> | 3 | 12.64 | 17.45 | 21.28 |
> | 4 | 12.67 | 17.43 | 21.26 |
> | 5 | 12.63 | 17.40 | 21.29 |
> | Avg | 12.628 | 17.416 | 21.274 |
> | Std | 0.0303315 | 0.03130495 | 0.01140175 |
>
> **Q3: Computational overhead.**
>
> Good suggestion!  We conduct a time analysis on VG, with training on the entire dataset and testing on 20 images. We report the mean value in **Table_R 9** of our ACC (w/o and with Step II in IGQS). We would like to claim that: 1) Our Step I in IGQS just introduces minor computational complexity in elementary matrix operations (cf. Eq. 1).  1) Due to the requirement of forward prediction for self-enhancement, Step II will induce extra computational overhead, but the performance gain brought by Step II is optional.
>
> **Table_R 9:  Inference costs on the VG dataset under the OVD+R SGG setting (Joint Base & Novel).**
>
> | Method | Training costs (min) |  Inference costs (s / I) | R@20 | R@50 | R@100 |
> | --- | --- | --- | --- | --- | --- |
> | OvSGTR (Swin-T) | 68 | 0.3871220016479492 | 10.02  | 13.50 | 16.37 |
> | ACC (Swin-T) | 71 | 0.3896771125793457 | 12.35 | 17.12 | 20.96 |
> | ACC (Swin-T) w/ Step II | 94 | 0.6402182579040527 | 12.61  | 17.43  | 21.27 |
>
> **Q4: More comparisons with recent baselines.**
>
> To address the reviewer's concern, we have included a comparison with a newly released method, OwSGG [3] (June 9, 2025), which adopts a multimodal prompting and embedding alignment strategy to enable pre-trained models such as LLaVA-Next and Qwen to generate scene graphs under an open-world setting.
>
> We report its performance on the VG dataset under two evaluation protocols: 1) the Open-vocabulary Relation (OvR) setting, and 2) the Open-vocabulary Detection + Relation (OvD+R) setting. The comparison results are shown below:
>
> **Table_R 10a: More experimental comparison results of OvR-SGG setting on VG test set.**
>
> | Method | Novel (Rel) R@50 | Novel (Rel) R@100 |
> | --- | --- | --- |
> | OwSGG (LLaVA-next) | 2.33 | 3.04 |
> | OwSGG (Qwen7b) | 1.15 | 1.67 |
> | OwSGG (Qwen72b) | 2.19 | 3.06 |
> | ACC (Swin-T) | 17.89 | 21.70 |
>
>  **Table_R 10b: More experimental comparison results of OvD+R-SGG setting on VG test set.**
>
> | Method | Novel (Obj) R@50 | Novel (Obj) R@100 | Novel (Obj) R@50 | Novel (Obj) R@100 |
> | --- | --- | --- | --- | --- |
> | OwSGG (LLaVA-next) | 2.37 | 3.07 | 2.33 | 3.04 |
> | OwSGG (Qwen7b) | 0.87 | 1.28 | 1.15 | 1.67 |
> | OwSGG (Qwen72b) | 1.88 | 2.73 | 2.19 | 3.06 |
> | ACC (Swin-T) | 17.16 | 21.10 | 15.90 | 19.46 |
>
> [3] Dutta, Amartya, et al. Open World Scene Graph Generation using Vision Language Models. CVPR Workshop 2025.
>
> **Q5: ACC vs. OvSGTR on out-of-distribution data.**
>
> Thank you for your valuable suggestion. We have tested both the OvSGTR and ACC models, trained on the VG dataset, on out-of-distribution datasets including PSG (**Table_R 11a**) and HICO-DET (**Table_R 11b**). The results are presented in the tables below:
>
> **Table_R 11a: Test on out-of-distribution PSG dataset.**
>
> | Method | R@20 | R@50 | R@100 |
> | --- | --- | --- | --- |
> | OvSGTR (Swin-T) | 11.30 | 13.87 | 15.81 |
> | ACC (Swin-T) | 12.10 | 15.19 | 17.23 |
>
> **Table_R 11b: Test on out-of-distribution HICO-DET dataset.**
>
> | Method | R@20 | R@50 | R@100 |
> | --- | --- | --- | --- |
> | OvSGTR (Swin-T) | 22.32 | 23.90 | 24.89 |
> | ACC (Swin-T) | 24.53 | 26.02 | 26.84 |
>
> As seen, our ACC demonstrates a striking performance advantage, significantly and consistently outperforming OvSGTR. This result provides strong evidence for the superior generalization capability of our approach, proving its effectiveness extends well beyond the original training domain. We will incorporate these compelling new results into the revised manuscript.
>
> **Q6: Key notations in Figure 2.**
>
> Thanks for your helpful suggestion. **Due to the limitation of rebuttal guidelines, we cannot include an external link / figure here.** We will add refined Figure 2 with key notations in the revision.
>
> **Q7: Typical failure scenarios.**
>
> Good suggestion! We analyze the examples where ACC misidentifies non-interacting object pairs, and find that:
>
> 1. For Interaction-Centric Knowledge Infusion, it is difficult to correctly match **small objects** and their related objects through bidirectional interaction prompts.
> 2. For Interaction-Centric Knowledge Transfer, when **multiple subject-object pairs with the same relational triplet categories** appear in the same image, the model might mistakenly match the subject in one triplet to the object in another triplet.
>
> **Since including an external link in the rebuttal is forbidden, we will add the visualization and discussion of failure cases in the revision.**
>
> **Q8: Minor points.**
>
> Thanks for your insightful comments. We will carefully refine the revision by:
>
> 1. Completing the sentence: "ACC achieves 29.28% R@100, which is higher than OvSGTR across both base and novel relations."
> 2. Adding quantitative results of Figure 5. We count the number of queries that match the corresponding GT's bbox (IoU > 0.5). The query quantities of matched GT's bbox in Figure 5 are 5 (original) *vs.* 10 (holding-guided) and 4 (original) *vs.* 11 (laying on-guided), respectively. It proves the effectiveness of our IGQS strategy in query initialization.
> 3. Correcting all reference formatting issues.

---

> > ### Comment · Reviewer_9mXA · 2025-08-06
> >
> > Dear authors,
> >
> > Thank you very much for the rebuttal. The rebuttal satisfactorily addresses several of my concerns, especially the answers to Q1, Q3, and Q5 are important and clearly strengthen the paper. I am not completely sold with the answer to Q4 -- the chosen baseline is so far off in terms of results that I am not sure this is the comparison the authors are looking for here.
> >
> > That said, I have a positive tendency on this paper. I am looking forward to further discussion, also with reviewer Y5Z7, who raised some interesting points.

---

> ### Author Response · Authors · 2025-08-07
> **Follow-up Response to Reviewer 9mXA**
>
> Sorry for the misunderstanding. Since the setting and splitting of OvD-R (with both novel objects and novel relations) were proposed by the recent OvSGTR (ECCV 2024), there are relatively few comparable baselines. The low performance result of OwSGG was **directly copied from their released paper** [3]. In addition, the baselines they compared (VS, OvSGTR, RAHP, cf. Table 3 in [3]) are ALL included in our paper (cf. Table 1 and Table 2).
>
> Furthermore, we reviewed the recently released papers and updated the more recent baselines (i.e., OpenSGen[4] and RTHP[5]) for comparison in Table_R 10. We hope the new comparison will satisfy you. Thank you!
>
> [4] Kong, Z. and Zhang, H., OpenSGen: Fine-Grained Relation-Aware Prompt for Open-Vocabulary Scene Graph Generation, ICMR 2025.
>
> [5] Feng, C., Xu, T., Wu, S., Xu, D. and Chen, E., Adaptive Hierarchical Prompt for Open-Vocabulary Scene Graph Generation, ACM Transactions on Asian and Low-Resource Language Information Processing 2025.
>
> **Table_R 10 More experimental comparison results of OvR-SGG setting on VG test set.**
>
> | Method | Venue | Base+Novel (Rel) R@50 | Base+Novel (Rel) R@100 | Novel (Rel) R@50 | Novel (Rel) R@100 |
> | --- | --- | --- | --- | --- | --- |
> | OwSGG (LLaVA-next) | CVPRW 2025 (June 9) | - | - | 2.33 | 3.04 |
> | OwSGG (Qwen7b) | CVPRW 2025 (June 9) | - | - | 1.15 | 1.67 |
> | OwSGG (Qwen72b) | CVPRW 2025 (June 9) | - | - | 2.19 | 3.06 |
> | OpenSGen | ICMR 2025 (June 30) | 18.0 | 20.5 | 15.7 | 17.9 |
> | RTHP | TALLIP 2025 (July 28) | 15.6 | 17.4 | - | - |
> | ACC (Swin-T) |  | 23.22 | 27.40  | 17.89  | 21.70 |

---

> > ### Comment · Reviewer_9mXA · 2025-08-08
> >
> > Thank you for these additional comparisons. Adding these would clearly strengthen the paper.

---

> > > ### Author Response · Authors · 2025-08-08
> > > **Follow-up Response to Reviewer 9mXA**
> > >
> > > We sincerely appreciate your further feedback. We will update these additional comparisons to the revision. We hope that our reply has addressed all your concerns. Thank you again!

---

### Official Review · Reviewer_Y5Z7 · 2025-07-05

**Clarity:** 3
**Significance:** 2
**Originality:** 1
**Rating:** 4
**Confidence:** 5

**Summary:**

This paper addresses the task of open-vocabulary scene graph generation by proposing a method that focuses on two key aspects: improving vision-language pre-training and enhancing query selection during fine-tuning.

**Interaction-Centric Knowledge Infusion**
- During the VLM pre-training phase, the authors introduce interaction-centric supervision by incorporating both traditional <subject–active verb–object> and novel <object–passive verb–subject> triplet forms.
- This is intended to better encode relational knowledge into the model.

**Interaction-Centric Knowledge Transfer**
- To address issues such as irrelevant query matching or matching objects without relationships (e.g., background noise), the method employs interaction-guided query selection.
- This ensures that only queries relevant to interactions are used for object detection.

**Interaction-Consistent Knowledge Distillation**
- Instead of aligning edge features directly, the paper proposes a relative-interaction retention distillation (RRD) strategy. RRD preserves the relative distances between edge features, encouraging consistency in the relational structure rather than absolute similarity.

**Questions:**

[Limited Technical Novelty]
- Is there a specific technical aspect of the proposed method that the authors believe is particularly novel or deserves emphasis?
- In the query selection module, previous works have also used objectness scores or similar heuristics for selecting relevant queries. To validate the effectiveness of the proposed interaction-centric approach, wouldn’t it be important to conduct ablation experiments using only object similarity (e.g., $\gamma=1$ in Eq. 1) or only interaction similarity ($\gamma=0$)? Such experiments would help isolate the contribution of each component.
- Also, what value of $\gamma$ was actually used in the experiments? I cannot find the value of $\gamma$ in both of the main paper and the supplementary and it is important for understanding the model behavior.

[Lack of Reasoning Behind Design Choices]
- Could the authors provide further justification or clarification for the design choices discussed above—particularly regarding the motivation for passive triplets in knowledge infusion and the decomposition strategy in query selection?

[Missing Reference]
- The formulation of the Relative-Interaction Retention Distillation (RRD) objective appears to be nearly identical to that of Relational Knowledge Distillation (RKD). I think it is necessary to refer the paper.

My main concern is that the technical novelty feels somewhat incremental, and the paper currently lacks sufficient reasoning to justify several of its core design decisions. I would appreciate a more thorough explanation to strengthen the motivation and clarity of the proposed approach.

**Ethical Concerns:**

["NO or VERY MINOR ethics concerns only"]

**Final Justification:**

The authors have addressed most of my concerns in the rebuttal. The paper is sufficiently novel and interesting, so I am raising my score by one level to borderline accept.

**Limitations:**

- The paper does not explicitly discuss the limitations of the proposed approach. Including a reflection on potential weaknesses—such as failure cases—would strengthen the paper.
- No ethical issues are apparent in the current work.

**Quality:**

3

**Strengths And Weaknesses:**

**Strengths**

[Well-Structured Paper]
- The paper is clearly written and easy to follow, with all mathematical formulations presented in a precise and understandable manner.
- The ablation study is well-designed and effectively demonstrates the contribution of each module to the overall performance.

[Novel Interaction-Centric Method]
- Unlike prior works that primarily rely on object-centric approaches, this paper introduces multiple interaction-centric techniques that improve performance across various components of the pipeline.
- The overarching motivation—infusing, transferring, and distilling interaction-aware knowledge—is coherently integrated into the proposed framework, resulting in a well-motivated and unified methodology.
- The authors successfully adapt existing techniques to the specific demands of the task by reformulating them in an interaction-centric manner, leading to a novel and cohesive overall architecture.
- In particular, the idea of applying interaction-centric principles to query selection is both intuitive and impactful.

**Weaknesses**

[Limited Technical Novelty]
- While the proposed methods are presented in an interaction-centric framework, many of the individual components appear to be only incremental extensions of existing techniques.
- In Interaction-Centric Knowledge Infusion, the bidirectional interaction prompt is essentially implemented by simply adding passive-voice triplets—e.g., <object–verb–subject>—which is a relatively straightforward modification.
- In Interaction-Centric Knowledge Transfer, the query selection mechanism builds on ideas that have already been widely explored in object detection literature [1, 2] or sementic segmentation literature [3].
- The Relative-Interaction Retention Distillation formulation is nearly identical to that of RKD (Relational Knowledge Distillation) [4], with no citation or significant deviation in formulation.
- While repurposing these techniques in an interaction-centric manner offers some novelty, the overall contribution feels somewhat incremental, especially given the lack of deeper reasoning behind these design choices (as discussed below).

[1] Y. Zhao, W. Lv, S. Xu, J. Wei, G. Wang, Q. Dang, Y. Liu, J. Chen, DETRs Beat YOLOs on Real-time Object Detection, CVPR 2024

[2] Y. Huang, H. Liu, H. Shuai, W. Cheng, DQ-DETR: DETR with Dynamic Query for Tiny Object Detection, ECCV 2024

[3] N. A. Shah, V. VS, V. M. Patel, LQMFormer: Language-aware Query Mask Transformer for Referring Image Segmentation, CVPR 2024

[4] W. Park, D. Kim, Y. Lu, M. Cho, Relational Knowledge Distillation, CVPR 2019

[Lack of Reasoning Behind Design Choices]
- In Interaction-Centric Knowledge Infusion
    - It's unclear why adding passive triplets (e.g., <surfboard–held by–man>) would reduce ambiguity.
    - If <man–hold–surfboard> is confusing, wouldn't its passive form be similarly ambiguous?
    - Since the two expressions encode essentially the same semantics, it's unclear how adding both to the embedding space provides a meaningful enhancement.

- In Interaction-Guided Query Selection,
    - The justification for decomposing triplets into <subject–predicate> and <predicate–object> pairs is not well-supported.
    - The paper claims that this helps reduce confusion (e.g., "man–riding" clarifies the ambiguity in "man–horse"), but the reasoning is hard to follow—especially since the decomposed pair may still retain ambiguity without full triplet context.

---

> ### Author Rebuttal · Authors · 2025-07-31
>
> We deeply appreciate reviewer Y5Z7 for the valuable time and constructive feedback. We provide point-to-point responses below.
>
> **Q1: Incremental extensions of existing techniques.**
>
> Sorry for the misunderstanding. Our core contributions lie in proposing a **unified interaction-centric framework**, designed specifically to solve the overlooked yet critical problem of “**relation pair mismatches**” in OVSGG. We would like to clarify two crucial techniques:
>
> 1. **Bidirectional Interaction Prompt in Interaction-Centric Knowledge Infusion.** Its novelty lies NOT in the linguistic/semantic transformation, but in **how it strategically leverages the attention mechanism of the grounding model's text encoder** (e.g., Grounding DINO). ****As mentioned in Line 187~190**,** in the passive form “surfboard held by man”, “surfboard” becomes the **syntactic subject**, thereby being more contextualized by “held”. We present the attention variance of the passive form and related discussion in **Q3**.
> 2. **Query Selection Mechanism in Interaction-Centric Knowledge Transfer**.  Since both OvSGTR and ACC are based on Grounding DINO, **language-guided query selection** (object-centric) is used as the default. This work further proposed IGQS, an **interaction-centric selection scheme** for OVSGG:
>     - **Objective:** Traditional query selection aims to improve general object detection or segmentation. Our IGQS is explicitly designed to resolve the **interaction ambiguity** among objects inherent in SGG.
>     - **Interaction Modeling:** We devise a unique two-step mechanism for modeling interaction. It first leverages both object and **relation** semantics for an initial ranking. More importantly, the second step is a self-reflective process that uses relations predicted from an initial pass to dynamically construct “interaction-pair” prompts. This interaction-focused query refinement is a novel design for the essential “mismatched” problem in OVSGG.
>
>     Furthermore, the ablation studies in Table 4 and the qualitative results in Figure 5 demonstrate the effectiveness of IGQS compared with the baseline (language-guided query selection in Grounding DINO). We provide more explanations of the IGQS design in **Q3**.
>
>     Nevertheless, we agree that IGQS’s query initialization draws high-level inspiration from the DETR-series in detection and segmentation. We will add relevant discussion in the revision. Thanks!
>
> **Q2: Missing citation of RKD.**
>
> Thanks for bringing this excellent work (i.e., RKD [2]) to our attention. We would like to clarify the main difference between RKD and our RRD:
>
> - **Motivation.** While RKD and our RRD use relationship distillation, they are motivated by different core problems. The original RKD was proposed as a general method for transferring knowledge by preserving the inter-sample relations of a teacher model's output. In contrast, our RRD is specifically motivated by a key challenge in OvSGTR, distinguishing meaningful **relational foregrounds** (true novel interactions) from a vast number of **background** object pairs. Our goal is to use relationship distillation as a targeted mechanism to help the model learn the structural differences between true interactions and random co-occurrences.
> - **Technique.** Our RRD is fundamentally different from RKD in that:  The distillation loss is computed on the set of **negative samples ($\mathcal{N}$)**. By aligning the structural similarity of these background triplets between the teacher and student, we explicitly encourage the student model to push the embeddings for **true, positive relations** away from the dense space of irrelevant **background** pairs.
>
> In addition, we conduct an experiment to empirically validate the advantage of our RRD. Concretely, we implemented RKD loss within our framework. As seen from **Table_R 3**, our RRD can get higher results compared with RKD. We will add these discussions and cite this work in the revision.
>
> **Table_R 3: Comparison of RKD [2] and RRD (ours) on the VG dataset under the OVD+R-SGG setting (Joint Base & Novel).**
>
> |Method|R@20|R@50|R@100|
> |-|-|-|-|
> |RKD|10.36|14.45|17.84|
> |RRD|11.43|15.67|19.20|
>
> [2] W. Park, D. Kim, Y. Lu, M. Cho. Relational Knowledge Distillation. CVPR 2019.
>
> **Q3: Deeper reasoning behind the design choices.**
>
> 1. **Passive triplets.** The benefit stems not from new semantic information but from the **asymmetry of the attention mechanism** within the text encoder (e.g., BERT). In the passive form “surfboard held by man”, “surfboard” becomes the syntactic subject, which **absorbs richer contextual semantics from the predicate** (e.g., “held”). This more discriminative textual feature is used to perform cross-attention with the visual patches from the image, which directly improves grounding accuracy. To empirically validate this, we display the attention scores of the active phrase (**Table_R 4a**) and the passive phrase (**Table_R 4b**). As seen, the subject in the passive phrase absorbs more from the predicate (”held”).
>
>     **Table_R 4a: Attention score across different tokens in “man hold surfboard”.**
>
>     |token | man | hold | surf | ##board |
>     |-|-|-|-|-|
>     |man|0.0891|**0.2062**|0.0307|0.0448|
>     |surf|0.0125|0.0160|0.0589|0.2721|
>     |##board|0.0227|0.0324|0.1365|0.0724|
>
>     **Table_R 4b: Attention score across different tokens in “surfboard held by man”.**
>
>     |token|surf|##board|held|by|man|
>     |-|-|-|-|-|-|
>     |man|0.0395|0.0352|0.0514|0.0495|0.0622|
>     |surf|0.0706|0.2585|0.0252|0.0396|0.0186|
>     |##board|0.1642|0.0768|**0.2096**|0.0363|0.0212|
>
> 2. **Decomposed strategy in query selection.** The core motivation for decomposing triplets is to **prevent the subject and object queries from interfering with each other** during the initialization phase, ensuring both are properly localized. When using a full <subject, predicate, object> prompt (e.g., “man hold racket”) to guide query selection, the initial queries concentrate on the **union box** of interacting pairs, leading to the **omission of relatively small objects within the union box**. To provide clear evidence, we visualize the query initialization results of the first image in Figure 5 under three different prompts (<subject, predicate, object>, <subject, predicate>, <predicate, object>), and count the number of queries that successfully matched GT boxes (IoU > 0.5).
>     - The composed prompt “man hold racket” led to queries focusing on the **union box** of the man and the racket (8 matched “man”, 3 matched “racket”).
>     - The decomposed subject-predicate prompt “man hold” focused queries on the **subject region** (10 matched “man”).
>     - The predicate-object prompt “hold racket” successfully focused queries on the **object region** (9 matched “racket”).
>
>     This analysis shows that our decomposed strategy is crucial for more accurate query initialization for both subjects and objects. **Since including an external link in the rebuttal is forbidden, we will add the visualization of query comparison in the revision.**
>
>     In addition, we conduct an experiment to study the effect of the decomposed strategy. As shown in **Table_R 5**, our decomposed strategy can lead to higher performance.
>
>
>     **Table_R 5: Ablation study on the effect of the decomposed strategy on the VG dataset under the OVD+R-SGG setting (Joint Base & Novel).**
>
>     | Decomposed Strategy | R@20 | R@50 | R@100 |
>     | --- | --- | --- | --- |
>     | ✕ | 10.89 | 15.17 | 18.72 |
>     | ✓ | 11.37 | 15.71  | 19.37 |
>
> **Q4: Technical novelty.**
>
> Our main novelty and contributions lie in proposing a **unified interaction-centric framework** designed to solve the overlooked problem of “**relation pair mismatches**” in OVSGG. The innovation is not just an isolated component, but a cohesive, complete solution that synergistically addresses this mismatch problem:
>
> - During the **knowledge infusion stage**, we enhance the quality of weak supervision signals through our bidirectional interaction prompts.
> - During the **knowledge transfer stage**, we devise interaction-guided query selection and interaction-consistent knowledge distillation to ensure the model focuses on true interactions while retaining its generalization capabilities during fine-tuning.
>
> This end-to-end, interaction-centric paradigm, from data generation to model optimization, represents our unique contribution to OVSGG.
>
> **Q5: Ablation study of object similarity and interaction similarity.**
>
> Thanks for the excellent suggestion. To study the contributions of object and relation similarities in our initial query selection, we conduct a detailed ablation study on the weight hyperparameter $γ$ (cf. Eq. 1). The results in **Table_R 6** clearly show that combining both similarities (0 < $γ$ < 1) outperforms using either component alone. Performance peaks at **$γ$**=0.7 (our default implementation), empirically validating our core hypothesis, i.e., incorporating relational context is critical for identifying the most relevant and interacting object candidates, leading to superior performance.
>
> **Table_R 6: Ablation study of similarity weight $γ$  on the VG dataset under the OVD+R-SGG setting (Joint Base & Novel).**
>
> |**$γ$**|R@20|R@50|R@100|
> |-|-|-|-|
> |0.0|10.02|13.50|16.37|
> |0.3|11.10|15.30|18.63|
> |0.5|11.21|15.43|19.04|
> |0.7|11.30|15.71|19.16|
> |1.0|10.68|14.17|17.83|
>
> **Q6: Limitations.**
>
> Thanks for your helpful suggestion. We discussed the limitations in **Appendix G:**
>
> *“it also inherits inductive biases from the teacher model. Like two sides of a coin, any biases in the vision-language model toward specific feature traits or classes may propagate to our model. Besides, our method can alleviate mismatched relational pairs, but cannot avoid all mismatches.”*
>
> We will add failure cases and discussion in the revision. For analysis of failure cases, please refer to **Q7** in **Reviewer E7mD.**

---

> > ### Author Response · Authors · 2025-08-07
> > **Open to Address Any Follow-up Concerns**
> >
> > Thank you once again for taking the time to review our work. We hope our response has addressed your concerns. As the discussion period is nearing its end, please don't hesitate to let us know if any remaining concerns require further discussion. We would be happy to provide any further clarification or elaboration.

---

> > ### Comment · Reviewer_Y5Z7 · 2025-08-07
> >
> > Most of my concerns have now been addressed, but a few issues remain:
> >
> > **[About Q2]**
> >
> > RKD already tries to align relational distances; the only difference is that RKD samples all pairs, whereas your paper uses only negative pairs. While the sampling strategy might be considered novel, the connection to RKD should be explicitly referenced.
> >
> > **[About Q3-1]**
> >
> > I still do not understand the need for introducing additional passive-voice verbs and think that there is some mismatch between motivation and the design. The Introduction (L49-54) states that knowledge infusion is necessary since pre-training supervision is noisy and produces triplet mismatches. According to the rebuttal and Section 3.1, however, the main change appears to be only a different attention pattern toward text features.
> >
> > If the model fails to accurately attend to the man who is holding the surfboard in “man hold surfboard,” there is no guarantee that it will correctly attend to the surfboard being held by that specific man in “surfboard held by man.” In other words, merely biasing attention toward the man in the first phrase and toward the surfboard in the second does not convince me that the model can reliably match which man is holding which surfboard, so the original mismatched triplet problem still seems unsolved.
> >
> > **[About Q3-2]**
> >
> > As mentioned, clearer visualizations of the relevant attention maps would be helpful.
> >
> > **[About Q5]**
> >
> > I strongly recommend that the related experiments be reported in the main paper.
> >
> > Apart from these points, most of my concerns have been resolved, and I am inclined to evaluate the paper positively.

---

> > > ### Author Response · Authors · 2025-08-09
> > > **Thanks for the Positive Evaluation**
> > >
> > > Thank you again for your time and insightful feedback throughout this discussion period! As the discussion will end in a few hours, we would like to know if our further response has addressed all your concerns.
> > >
> > > If you have any further questions, please feel free to reach out. We are happy to provide additional clarifications.
> > >
> > > Gentle Reminder: We are pleased to hear you have a **positive evaluation** of our paper. We would sincerely appreciate it if you could kindly consider **reflecting this in the final rating**.

---

> ### Comment · Area_Chair_4LJq · 2025-08-05
>
> Dear Reviewer Y5Z7,
>
> The author-rebuttal phase is now underway, and the authors have provided additional clarifications and performance results in their rebuttal. Could you please take a moment to review their response and engage in the discussion? In particular, we’d appreciate your thoughts on whether their revisions adequately address your initial concerns. Thank you for your time and valuable contributions.
>
> Best, Your AC

---

> ### Author Response · Authors · 2025-08-08
> **Follow-up Response to Reviewer Y5Z7**
>
> Thank you for your valuable time and feedback! We would address your remaining concerns as follows. We hope our follow-up responses have fully addressed your concerns.
>
> For **Q1**, in the first response, we have committed to explicitly citing this paper. Once again, we thank the reviewer for bringing this outstanding work to our attention.
>
> For **Q3-1**, thank you for your further illustration of your concern. We would like to clarify that the overall intention of Bidirectional Interaction Prompt is to reduce mismatch, but passive-voice is designed to enable more interacting objects to be detected (relevant discussions have been given **Sec.** **F.1 in the Appendix**).
>
> - ***Problem with the Baseline ("Object Prompt").*** In this paper, we argue that the **original object prompt** (i.e., directly concatenating the subject & object entities, e.g., “man. surfboard.”) generates **redundant object boxes, easily causing mismatches in associating object pairs and noisy pseudo supervision** (cf. L49-54 and Figure 1)**.**
> - ***Design 1. Adding Interaction Context ("Interaction Prompt").*** **Interaction Prompt** can **alleviate the mismatches** by **modeling interaction** information to distinguish interacting objects (e.g., man involved in holding action) from non-interacting ones through the attention mechanism. (cf. L68-68, L184-187, and L965-970)
> - ***Design 2. Augment the object’s role with Passive-voice ("Bidirectional Interaction Prompt").***  While interaction prompt significantly **reduces the number of** **redundant object boxes**, it often over-focuses on the subject (e.g., detecting only the "man" subject bounding box), leading to the omission of critical object boxes (cf. **Figure S3 in the Appendix**). That is **the reason for introducing the passive-voice verbs to augment the object role** (cf. L187-190 and L973-977 and rebuttal).
> - ***Rule-based combination.*** After the boxes for the interacting subject/object were detected, we adopted a rule-based combination to enable matches with IoU overlap (cf. L191-192).
>
> To empirically validate this, we have conducted experiments to compare three prompts in Table_R 14. As the results clearly show, each component of our design provides a significant boost.
>
> Admittedly, the model cannot accurately attend to ALL objects in prompts.  Nevertheless, we do our best to ensure that more interacting object pairs can be detected and matched.
>
> **Table_R 14. Comparison of different pre-training prompt strategies, evaluated directly on the VG150 test set.**
>
> | Prompt | R@20  | R@50  | R@100 |
> | --- | --- | --- | --- |
> | **Object Prompt** | 6.61  | 8.92  | 10.90 |
> | **Interaction Prompt (wo passive-voice)** | 7.35 | 10.06 | 12.27 |
> | **Bidirectional Interaction Prompt (w/ passive-voice)** | 7.86  | 10.81 |  13.31 |
>
> For **Q3-2**,  as said, since including **an external link (Figure) in the rebuttal is forbidden**, we will add the visualization in the revision.
>
> For **Q5**, we will add the reported experiments in the main paper of the revision.

---

### Official Review · Reviewer_Ae9W · 2025-07-08

**Clarity:** 3
**Significance:** 3
**Originality:** 3
**Rating:** 4
**Confidence:** 4

**Summary:**

The paper proposes an interaction-centric end-to-end OVSGG framework to reduce the pervasive mismatch between interacting/non-interacting object pairs. The paper introduces a bidirectional interaction prompt to facilitate visual triplet pseudo-supervision generation, therefore achieving interaction-centric knowledge infusion. Besides, it constructs interaction-guided query selection and incorporates interaction-consistent KD to achieve the interaction-centric knowledge transfer.

**Questions:**

Please refer to the weaknesses.

**Ethical Concerns:**

["NO or VERY MINOR ethics concerns only"]

**Final Justification:**

After reading all the reviews and responses, I maintain my rating score.

**Limitations:**

Yes.

**Quality:**

3

**Strengths And Weaknesses:**

Strengths:
1. The motivation is reasonable. The problem claimed in the paper is a key challenge in scene graph generation.
2. The paper constructed a transfer-learning-based framework, which is suitable for an open vocabulary setting.


Weaknesses:
1. Although the paper focuses on open-vocabulary scene graph generation, the evaluation on base relation should be included.
2. Considering that the paper focuses on action-centric scene graph exploration, the authors should include a comparison with the HOI task, which focuses on human interactions.
3. The motivation claimed in the abstract has little relationship with open vocabulary setting; the authors should claim that the challenge existed in open vocabulary scene graph generation.

---

> ### Author Rebuttal · Authors · 2025-07-31
>
> We deeply appreciate reviewer Ae9W for the valuable time and constructive feedback. We provide point-to-point responses below.
>
> **Q1:  Evaluation on the base relation.**
>
> As suggested, we have added the evaluation on **base object and relation categories** in **Table_R 1**. The results clearly show that the proposed ACC significantly outperforms previous SOTA (i.e., OvSGTR) on both novel and base classes. This demonstrates that our approach provides a more comprehensive and powerful generalization capability, enhancing performance across the board, not just for unseen classes. We will incorporate these metrics into the revised manuscript.
>
> **Table_R 1: Experimental results of OvD+R-SGG setting on VG test set.**
>
> | Method | Joint Novel + Base R@20/50/100 |  Base (Obj) R@20/50/100  | Novel (Obj) R@20/50/100 | Base (Rel) R@20/50/100 | Novel (Rel) R@20/50/100 |
> | --- | --- | --- | ---- | ------- | --- |
> | OvSGTR (Swin-T) | 10.02 / 13.50 / 16.37 | 8.78 / 11.95 / 14.79 | 10.56 / 14.32 / 17.48 | 12.07 / 16.47 / 20.09 | 7.09 / 9.19 / 11.18 |
> | ACC (Swin-T) | 12.61 / 17.43 / 21.27 | 11.66 / 16.46 / 20.35 | 12.48 / 17.16 / 21.10 | 12.20 / 16.67 / 20.57 | 11.38 / 15.90 / 19.46 |
>
> **Q2: Comparison with the HOI task.**
>
> Thanks for your valuable feedback. We agree that a comparison with the HOI detection task provides valuable context for our work. We will incorporate a detailed discussion clarifying the distinctions between these two tasks in the revised manuscript. In brief, the key difference lies in their scope and objective. **HOI detection**, particularly on benchmarks like HICO-DET [1], is primarily a detection task over a predefined, closed set of specific human-centric interactions (e.g., 600 <action, object> pairs). In contrast, **SGG** addresses a more general and compositional challenge: generating <subject, action, object> triplets between ***any pair*** of objects, making the potential output space combinatorially large (e.g., 150x50x150 potential triplets for VG).
>
> Furthermore, to empirically validate the effectiveness of the proposed ACC, we evaluate ACC and OvSGTR on the HICO-DET benchmark. As shown in **Table_R 2**, ACC consistently outperforms OvSGTR, achieving a **2.54**% absolute improvement in R@100 of novel classes. This result is significant: it demonstrates that our model's core principles are so robust. They excel not only on the general OVSGG task they were designed for, but also on the specialized HOI task. We will incorporate these new compelling results into the revised manuscript.
>
> **Table_R 2: Performance comparison on the HICO-DET [1] dataset under the OvR-SGG setting.**
>
> | Method | Novel + Base R@20/50/100 | Base R@20/50/100 | Novel R@20/50/100 |
> | --- | --- | --- | --- |
> | OvSGTR (Swin-T) | 34.62 / 37.39 / 39.04 | 36.87 / 38.12 / 40.51 | 22.94 / 28.48 / 31.84 |
> | ACC (Swin-T) | 35.74 / 38.58 / 40.19 | 37.70 / 40.11 / 41.35 | 24.44 / 30.77 / 34.38 |
>
> ---
>
> [1] Chao YW, Liu Y, Liu X, Zeng H, Deng J. Learning to detect human-object interactions. WACV 2018.
>
> **Q3: Claim challenges in OVSGG in Abstract.**
>
> Good suggestion! We will revise the abstract to clearly highlight the challenges in OVSGG. Below is the updated abstract, with bold text indicating the revised parts:
>
> *Open-vocabulary scene graph generation (OVSGG) extends traditional SGG by recognizing novel objects and relationships beyond predefined categories, leveraging the knowledge from pre-trained large-scale models. Existing OVSGG methods always adopt a two-stage pipeline: 1) Infusing knowledge into large-scale models via pre-training on large datasets; 2) Transferring knowledge from pre-trained models with fully annotated scene graphs during supervised fine-tuning. However, due to a lack of explicit interaction modeling, these methods struggle to distinguish between interacting and non-interacting instances of the same object category.* ***This limitation induces critical issues in both stages of OVSGG: it generates noisy pseudo-supervision from mismatched objects during knowledge infusion, and causes ambiguous query matching during knowledge transfer*.**

---

> > ### Author Response · Authors · 2025-08-07
> > **Open to Address Any Follow-up Concern**
> >
> > Thank you once again for taking the time to review our work. We hope our response has addressed your concerns. As the discussion period is nearing its end, please don't hesitate to let us know if any remaining concerns require further discussion. We would be happy to provide any further clarification or elaboration.

---

> ### Comment · Area_Chair_4LJq · 2025-08-05
>
> Dear Reviewer Ae9W,
>
> The author-rebuttal phase is now underway, and the authors have provided additional clarifications and performance results in their rebuttal. Could you please take a moment to review their response and engage in the discussion? In particular, we’d appreciate your thoughts on whether their revisions adequately address your initial concerns. Thank you for your time and valuable contributions.
>
> Best, Your AC

---

### Note · Authors · 2025-08-12

We sincerely appreciate all reviewers for their constructive feedback. We are encouraged that they found our work **well-motivated** [9mXA, Y5Z7], addressing a **key challenge** [Ae9W], and our interaction-centric idea to be **novel** [Y5Z7, E7mD], **intuitive** [Y5Z7], **technically sound** [E7mD], and **simple yet effective** [9mXA]. Meanwhile, we are glad they think the paper is **well-structured** [Y5Z7], **very well written** [9mXA], and **easy to follow** [Y5Z7]. Additionally, our **well-designed** [Y5Z7] experiments and **careful** [9mXA] ablations yield **promising** [9mXA, E7mD] results that **significantly and** **consistently outperform** [9mXA] baselines. We are pleased that reviewers recognize our approach as a **fundamental technique with the potential for universal grounding enhancement** [E7mD]. The major changes in our revision are summarized as follows:
- For Reviewer Ae9W:
    - Evaluation.  We present results on base object /relation categories.
    - Comparison. We add a comparison on the HICO-DET benchmark.
    - Presentation. We refine the abstract to highlight the challenges in OVSGG.
- For Reviewer Y5Z7:
    - Clarification. We provide the deeper reasoning for our design choices, supported by attention score analysis and an ablation on the query selection strategy.
    - Citation. We explicitly cite and discuss the RKD.
    - Experiment. We add an ablation on the similarity weight.
    - Visualization. We include attention maps and query visualizations.
    - Discussion. We expand the limitations section with a discussion of failure cases.
- For Reviewer 9mXA:
    - Metric. We add the mR@K metric to the main tables.
    - Experiment:  We add experiments on OOD datasets. We also compare ACC with recent baselines (OwSGG, OpenSGen, and RTHP).
    - Analysis. We report standard deviations from repeated experiments. We also add an analysis of the computational overhead. Besides, we offer the quantitative analysis of Figure 5.
    - Visualization. We refine Figure 2 with key notations. Additionally, we add the visualizations of failure scenarios and relevant discussion.
    - Presentation. We correct all typos and reference formatting.
- For Reviewer E7mD:
    - Analysis. We add a generalization study of bidirectional prompts on object detection. We also offer a detailed analysis of training with fully-annotated SGG data.
    - Clarification. We offer a clear clarification of why the KD is critical for OVSGG.

Sincerely yours,

Authors.

---

### Decision · Program_Chairs · 2025-09-17

**Decision:**

Accept (poster)

**Comment:**

This paper presents an interaction-centric open-vocabulary scene graph generation (OVSGG) framework to better handle the pervasive mismatch between interacting and non-interacting object pairs. It introduces three key modules: bidirectional interaction prompts, interaction-guided query selection, and interaction-consistent knowledge distillation, showing improved performance on novel objects and relationships.

The paper received mixed reviews with clear strengths and weaknesses. The reviewers appreciated the novel interaction-centric approach to open vocabulary scene graph generation, achieving promising experimental results that consistently improved over the baselines. The clear presentation, well-motivated approach, and unified framework with effective ablation studies were highlighted as positives.
However, the reviewers raised significant concerns about: 1) limited technical novelty, as many components appear to be incremental extensions of existing techniques without proper citations, particularly the Relative-Interaction Retention Distillation; 2) insufficient evaluation, including missing metrics (mR@K despite claims), absence of standard deviations, limited comparison with recent baselines, and no out-of-distribution testing; 3) lack of clear reasoning behind key design choices, such as why adding passive triplets or implementing student model distillation; 4) no discussion of computational overhead.

During the author-reviewer discussion, the authors provided a detailed response including further clarification on design choices and novelty relative to related work, additional evaluations with mR@K metrics, comparisons to recent methods, OOD testing, and analysis of computational overhead. The rebuttal addressed most of the reviewers' concerns, and three reviewers responded positively to the discussion. Specifically, Reviewers 9mXA and E7mD recommended acceptance, while the other two reviewers maintained a positive assessment, conditioned on including the additional information from the rebuttal.

The AC largely concurs with the reviewers' assessment: the paper introduces an interesting and effective approach to OVSGG, and its merits outweigh its weaknesses. The AC also notes that while the added experiments and clarifications from the rebuttal stage are substantial, they can be incorporated in revision. Therefore, the AC recommends acceptance. The authors should revise the manuscript to address the reviewers' feedback and incorporate the points discussed in the rebuttal.